# Mirostat: A Neural Text Decoding Algorithm that Directly Controls Perplexity

**Sourya Basu**[*]     **Govardana Sachitanandam Ramachandran**[†]     **Nitish Shirish Keskar**[†]

**Lav R. Varshney**[*,†]
[*]Department of Electrical and Computer Engineering, University of Illinois at Urbana-Champaign
[†]Salesforce Research

## Abstract

Neural text decoding algorithms strongly influence the quality of texts generated using language models, but popular algorithms like top-$k$, top-$p$ (nucleus), and temperature-based sampling may yield texts that have objectionable repetition or incoherence. Although these methods generate high-quality text after ad hoc parameter tuning that depends on the language model and the length of generated text, not much is known about the control they provide over the statistics of the output. This is important, however, since recent reports show that humans prefer when perplexity is neither too much nor too little and since we experimentally show that cross-entropy (log of perplexity) has a near-linear relation with repetition. First we provide a theoretical analysis of perplexity in top-$k$, top-$p$, and temperature sampling, under Zipfian statistics. Then, we use this analysis to design a feedback-based adaptive top-$k$ text decoding algorithm called *mirostat* that generates text (of any length) with a predetermined target value of perplexity without any tuning. Experiments show that for low values of $k$ and $p$, perplexity drops significantly with generated text length and leads to excessive repetitions (the boredom trap). Contrarily, for large values of $k$ and $p$, perplexity increases with generated text length and leads to incoherence (confusion trap). Mirostat avoids both traps. Specifically, we show that setting target perplexity value beyond a threshold yields negligible sentence-level repetitions. Experiments with human raters for fluency, coherence, and quality further verify our findings.

## 1 Introduction

Large-scale generative language models (LMs) have received recent attention due to their high-quality open-ended text generation ability (Brown et al., 2020; Radford et al., 2019). Generating texts from these LMs usually relies on some form of random sampling. Pure sampling often leads to incoherent and low-quality texts (Holtzman et al., 2018), whereas greedy decoding leads to excessive repetitions, another form of low quality. The right decoding algorithm is needed to generate high-quality texts with controlled attributes (Ippolito et al., 2020; Zhang et al., 2020; Ippolito et al., 2019).

We introduce mirostat,[1] a neural text decoding algorithm that *actively controls* the generative process to maintain the perplexity of generated text at a certain desired value. Mirostat uses an adaptive top-$k$ sampling algorithm to actively tune the value of $k$ which helps maintain the overall perplexity of the text; recall that top-$k$ sampling (Holtzman et al., 2018; Fan et al., 2018) is where the next word is sampled from the top $k$ most probable choices.

Top-$k$ sampling and several other recent sampling methods are motivated by suppressing an unreliable tail in the probability distribution of trained LMs. Another sampling method is top-$p$, also known as *nucleus sampling*, where the next word is chosen from the top $x$ probable choices, where

---

[1]The word mirostat is derived from *mirum* which is Latin for *surprise* and *stat* meaning control.

This work was funded in part by the IBM-Illinois Center for Cognitive Computing Systems Research (C3SR), a research collaboration as part of the IBM AI Horizons Network and the National Science Foundation Grant CCF-1717530.

$x$ is the smallest integer such that their cumulative probability mass is at least $p$ (Holtzman et al., 2020). While top-$k$ sampling involves a fixed number of most probable choices, top-$p$ sampling involves a dynamic number of choices based on a fixed $p$ value and shows better statistical and human-evaluated performance. For small values of $k$ and $p$, these sampling methods unfortunately repeat phrases in generated text. This can be handled by penalizing repetitions and using appropriate *temperature* values (Keskar et al., 2019) or adding diversity to the generated text (Zhang et al., 2020; Vijayakumar et al., 2018). On the other hand, large values of $k$ and $p$ can lead to incoherent texts similar to pure sampling. Although choosing appropriate values of $p$ or $k$ can avoid repetition and incoherence, this involves ad hoc tuning of parameters. Even for a fixed value of $p$ or $k$, the generated text can have varying statistical properties.

Intriguingly, as we demonstrate via Example 1 in Appendix A, small values of a certain perplexity statistic of generated texts called *surprise* (Def. 1) are closely linked to repetitions and large values of surprise are linked to incoherence. Perplexity is a statistical metric used to evaluate quality of neural text generation, and is closely related to average surprise as shown in Fig. 7 in Appendix A and formalized in Sec. 2. A large-scale human subject experiment by Zhang et al. (2020) showed human-evaluated quality is closely related to the likelihood of the generated text for fixed number of tokens. In particular, reducing perplexity increases quality upto some point before the quality starts dropping. This implies that good control over perplexity of the generated text would give direct control over the quality of generated text (as evaluated by humans). Generating texts with an appropriately chosen target perplexity value may maximize quality of generated text. *Ergo mirostat*.

Now we summarize our key contributions. Sec. 3 shows theoretically how cross-entropy and hence perplexity grows in top-$k$ and top-$p$ sampling as a function of $k$ and $p$ respectively, which was previously unknown. Sec. 4 introduces mirostat sampling, which outputs texts with predetermined target perplexity value. Although perplexity may not fully capture the quality of text (Hashimoto et al., 2019), much literature discusses its correlation to quality (Zhang et al., 2020). Hence, our algorithm to control perplexity helps generate high-quality text. Sec. 5.1 experimentally shows much fluctuation in cross-entropy rates in top-$k$ and top-$p$ sampling as a function of their input parameters, hence unable to control perplexity of output text. Sec. 5.2 shows repetition is closely related to perplexity of the generated texts, mostly independent of the sampling method, but slightly dependent on the LM used. Sec. 5.3 experimentally shows mirostat sampling avoids both boredom and confusion traps for a wide range of target perplexity values. Sec. 5.4 provides our own experiments with human raters that demonstrate mirostat efficacy for fluency, coherence, and overall quality.

## 1.1 RELATED WORK

**Sampling from distorted probability distribution**    Pure sampling from LMs often leads to incoherent text whereas greedy decoding leads to repetitions. Distorting probability distributions, as in top-$k$, top-$p$, or temperature sampling help improve quality of generated texts, if parameters are properly tuned (Holtzman et al., 2018; Fan et al., 2018; Holtzman et al., 2020). Tuning these methods, however, is ad hoc and does not provide good control over the statistics of the output. Our method uses statistics of previously-generated tokens as input to generate the next token, by distorting the probability distribution so it helps control the overall statistics of the generated text. This ability to control the perplexity of the output is a key advantage of our method over previous work. This, when used with the relation between perplexity and human-evaluated quality observed by Zhang et al. (2020), can yield text that has better quality control.

**Controllable text generation**    Controllable text generation has oft focused on semantics of the output text, as in LMs like CTRL (Keskar et al., 2019), and sampling algorithms like plug-and-play LM (Dathathri et al., 2020) and constrained sentence generation by Metropolis-Hastings (Miao et al., 2019). Contrarily our approach is purely statistical, guiding the decoder along a desired statistical path that addresses issues with pure sampling and greedy decoding.

**Quality-diversity tradeoff**    Top-$k$, top-$p$, and low-temperature sampling improve the quality of the text, but at the cost of reduced diversity. Applications like question-answering only demand high-quality generation, but open-ended tasks such as story generation demand diversity too. Li et al. (2016); Vijayakumar et al. (2018); Kulikov et al. (2019) propose variants of beam search to induce diversity in generated text. However, Zhang et al. (2020) observe a tradeoff between quality and

diversity; they further observe diversity is closely related to entropy whereas quality is maximized in a certain range of observed likelihood values for fixed-length sentences. Our algorithm well-controls observed cross-entropy, the observed likelihood per token of generated text. Hence, by maintaining the observed cross-entropy in a certain range, we can ensure high-quality text generation.

**Repetitions**  Greedy decoding from LMs often lead to texts with excessive repetitions both at token- and sentence-levels. Several techniques have been proposed to address this. Token loss dynamic reweighting (TLDR) hypothesizes some tokens are more difficult to learn than others and so reweighting tokens during learning can balance things to reduce repetitions (Jiang et al., 2020). Keskar et al. (2019) use a repetition penalty in decoding to reduce repetition of tokens. Welleck et al. (2020) suggest the cause for repetitions is a flaw in the training objective itself and use a new objective that gives less probability to unlikely sequence including texts with high repetitions. Variants of top-$k$ sampling and repetition penalty in (Keskar et al., 2019) were used before by Foster & White (2007) to reduce repetitions. Here, we demonstrate a near-linear relation between repetitions and observed cross-entropy and so we directly control repetitions by controlling observed cross-entropy.

## 2   SURPRISE, CROSS-ENTROPY, AND PERPLEXITY

Here we formally define surprise, cross-entropy, and perplexity. For a random variable $X \in \mathcal{X}$ distributed as $P$, the surprisal associated with an instance $x$ of $X$ is defined as $-\log P(x)$ (Han & Kobayashi, 2007). Hence, less probable instances are more surprising than more probable instances. Extending the definition to conditional random variables, we next define the surprise associated with tokens and sentences with respect to generated text for a fixed model distribution $P_M$.

**Definition 1.** The *surprise* value of a token $X$ with respect to generated text $X_{<i}$ and model distribution $P_M$ for some fixed model $M$ is $\mathfrak{S}_M(X|X_{<i}) = -\log P_M(X|X_{<i})$.

We will soon see this quantity is directly related to perplexity. Now we define the average surprise for a sentence $X$ with $n$ tokens.

**Definition 2.** For a sentence $X^n = (X_1, \dots, X_n)$ with $n$ tokens, the *surprise rate* with respect to a probability distribution $P_M$ for some model $M$ is $\overline{\mathfrak{S}}_M(X^n) = -\frac{1}{n} \sum_{i=1}^{n} \log P_M(X_i|X_{<i})$.

The cross-entropy of a discrete random variable $X \in \mathcal{X}$ distributed as $P_M$ with respect to a discrete random variable $Y \in \mathcal{Y}$ distributed as $P_N$ such that $\mathcal{Y} \subseteq \mathcal{X}$ is $H(P_N, P_M) = -\sum_{y \in \mathcal{Y}} P_N(y) \log P_M(y) = \mathbb{E}_{P_N}[\mathfrak{S}_M(Y)]$. The cross-entropy rate of a stochastic process $\mathcal{X} = \{X_i\}, X_i \in \mathcal{X}$ distributed as $P_M$ with respect to a stochastic process $\mathcal{Y} = \{Y_i\}, Y_i \in \mathcal{Y}$ distributed as $P_N$ and $\mathcal{Y} \subseteq \mathcal{X}$ is defined as $\mathcal{H}(P_N, P_M) = \lim_{n \to \infty} \mathbb{E}_{P_N}[\overline{\mathfrak{S}}_M(Y^n)]$, when the limit exists. Further, if $Y^n$ is sampled from $P_N$ and if $P_N$ is a stationary ergodic source, then by the Shannon-McMillan-Breiman theorem (Cover & Thomas, 2006, Thm. 16.8.1), we have $\lim_{n \to \infty} \overline{\mathfrak{S}}_M(Y^n) = \mathcal{H}(P_N, P_M)$, when the limit exists. Now, the perplexity corresponding to $\mathcal{H}(P_N, P_M)$ is simply $\text{PPL}(P_N, P_M) = 2^{\mathcal{H}(P_N, P_M)}$, following Brown et al. (1992); Varshney et al. (2020). For experiments, when the text is generated using $P_N$, we approximate $\mathcal{H}(P_N, P_M)$ by $\overline{\mathfrak{S}}_M(Y^n)$ for a sentence of length of $n$. This is because natural language shows stationary ergodic property (Manning & Schutze, 1999). Perplexity denotes how close $P_N$ is to $P_M$. The lower the perplexity, the closer the distributions $P_N$ and $P_M$.

## 3   THEORETICAL ANALYSIS OF SAMPLING METHODS

Here we summarize theoretical results for different sampling methods; details and proofs in App. B.

Zipf's law states that the frequency of occurrence of any word in the vocabulary is inversely proportional to its rank in the frequency table (Zipf, 1965; Powers, 1998). More precisely, for a vocabulary of size $N = |\mathcal{V}|$ the frequency of the $i$th most probable word is

$$p(i; s, N) = 1/(i^s H_{N,s}), \tag{1}$$

where $s$ is an exponent characterizing the distribution and $H_{N,s} = \sum_{n=1}^{N} \frac{1}{n^s}$ is the $N$th generalized harmonic number. Further, for human languages the exponent $s$ is very close to 1. Hence, when

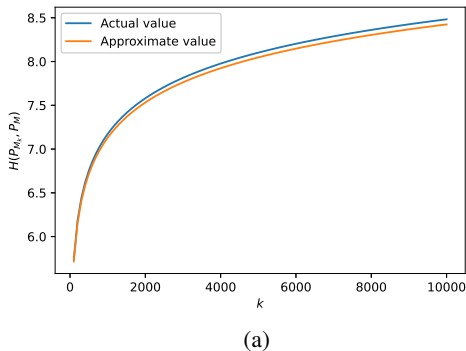 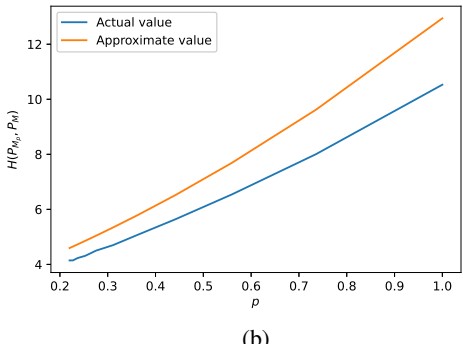

|  (a) | (b) |

Figure 1: Theoretical analysis of cross-entropy in top-$k$ and top-$p$ sampling: (a) approximation to $H(P_{M_k}, P_M)$ obtained in Thm. 2, (b) approximation to $H(P_{M_p}, P_M)$ obtained in Thm. 4 for $s = 1.1$, $N = 50,000$. Note that $H(P_{M_k}, P_M)$ grows sharply for small values of $k$ and the growth is negligible for large values of $k$, whereas $H(P_{M_p}, P_M)$ grows nearly linearly with $p$.

required, we write $s = 1 + \epsilon$, for some small $\epsilon > 0$. For all of our theoretical analysis, we assume the sampled words follow Zipf's law.

First we summarize results for top-$k$ sampling. Thm. 1 shows that $\mathfrak{S}(k)$ grows steeply for small values of $k$, but grows very slowly for large values of $k$. Thm. 2 computes an approximation for $H(P_{M_k}, P_M)$; Fig. 1a shows this approximation is very good. Since $H(P_{M_k}, P_M)$ does not grow much beyond $k = 2000$, it makes sense to tune $k$ between 1 to 2000 to get a desired cross-entropy.

Now we summarize the results for top-$p$ sampling. Thm. 3 proves that $\mathfrak{S}(p)$ behaves near-linearly in $p$. Further, Thm. 4 provides approximate expressions for $H(P_{M_p}, P_M)$ that show $H(P_{M_p}, P_M)$ grows approximately linearly with $p$; this approximate linearity is also shown in Fig. 1b. This is in contrast to top-$k$ sampling where $H(P_{M_k}, P_M)$ is highly nonlinear. Temperature is used to suitably distort the original distribution in so as to generate samples that avoid problems associated with pure sampling. In particular, lowering the temperature makes the sampling more greedy. For a given temperature $T > 0$, the frequency of the $k$th most probable word in (1) is given by $p(k; s, N, T) = 1/\left(k^{\frac{s}{T}} H_{N,\frac{s}{T}}\right) = p(k; \frac{s}{T}, N)$. Hence the effect of temperature in our analysis is captured simply by modifying $s$ to $s/T$.

## 4 PERPLEXITY-CONTROLLED TEXT GENERATION

In this section we propose the mirostat algorithm[2] to directly control the cross-entropy rate of the generated text. Mirostat works in two stages for generating each word. First it estimates the value of $s$ assuming words follow Zipf's law, details of which is given in Appendix C. Then, it uses top-$k$ sampling where $k$ is a function of the estimated $s$ and of the target surprise value of the output text.

Alg. 1 details mirostat,which generates texts with predetermined average surprise value. The input is a target surprise value $\tau$, which in turn initializes a variable $\mu = 2\tau$. Each word is sampled by first estimating $s$ from (30) as $\hat{s}$, then using top-$k$ sampling by approximating $k$ as a function of $\hat{s}$ and $\mu$ according to $H_{N,s} \approx \int_1^N \frac{1}{t^s} dt = \frac{(1-N^{s-1})}{s-1}$ and using (3) to get, with $\hat{\epsilon} = \hat{s} - 1$,

$$k = \left((\hat{\epsilon}2^\mu)/(1 - N^{-\hat{\epsilon}})\right)^{\frac{1}{\hat{s}}}, \tag{2}$$

We initialize $k$ corresponding to surprise value $2\tau$ and not $\tau$ since we are sampling from top-$k$ and not computing the surprise value at $k$ itself. This $2\tau$ initialization works well in practice, and the rest is taken care of by feedback: An error term $e$ is computed as the difference between the observed surprise $\mathfrak{S}(X)$ of the sampled word $X$ and $\tau$, which then is used to update $\mu$. Note that we can use an alternate algorithm to tune $k$ in Alg. 1 by iterating through the most probable tokens to set $k$

---

[2]Code is available at https://github.com/basusourya/mirostat

corresponding to a token that has a suitable amount of surprise. More details on such an algorithm with experimental results are given in Appendix E.

---

**Algorithm 1:** Mirostat sampling for perplexity control

---

Target cross-entropy $\tau$, maximum cross-entropy $\mu = 2\tau$, learning rate $\eta$, $m = 100$
**while** *more words are to be generated* **do**
  Compute $\hat{s}$ from (30): $\hat{s} = \sum_{i=1}^{m-1} t_i b_i / \sum_{i=1}^{m-1} t_i^2$
  Compute $k$ from (2): $k = \left( \hat{\epsilon} 2^\mu / (1 - N^{-\hat{\epsilon}}) \right)^{(1/\hat{s})}$
  Sample the next word $X$ using top-$k$ sampling
  Compute error: $e = \mathfrak{S}(X) - \tau$
  Update $\mu$: $\mu = \mu - \eta e$
**end**

---

## 5 EXPERIMENTAL ANALYSIS

Here we provide experiments for the performance of top-$k$, top-$p$, and mirostat sampling. We use the GPT-2 LM with 117M parameters for all experiments (Radford et al., 2019) unless mentioned otherwise, and just refer to it as GPT-2. One main takeaway is that unlike other approaches, mirostat indeed provides direct control over the observed cross-entropy of the output text.

### 5.1 CROSS-ENTROPY RATE FOR DIFFERENT SAMPLING METHODS

Fig. 2 plots observed cross-entropy in generated texts versus several input parameters for different sampling methods. For each plot, we generate four output texts of 200 tokens corresponding to each value of input parameter in each sampling method with same context in each case.

Fig. 2a shows the observed surprise values in generated texts versus $k$ in top-$k$ sampling. Note that cross-entropy has a steep increase for small values of $k$ and relatively slow increase in $k$ for high values of $k$. Thus, for small values of $k$, cross-entropy is very sensitive to change in the $k$, but, for large values of $k$, cross-entropy hardly changes. Even though we can clearly see the increase in cross-entropy with increase in $k$, it is difficult to control cross-entropy using top-$k$ sampling.

Fig. 2b plots the observed surprise values in generated text versus $p$ in top-$p$ sampling. Observe that cross-entropy grow essentially linearly with increase in $p$ unlike top-$k$ sampling.

Fig. 2c plots the observed cross-entropy in generated texts versus target cross-entropy in mirostat sampling, Alg. 1. Observe that mirostat sampling gives very good control over observed surprise value, with low variance for surprise values less than five. For higher target surprise, Alg. 1 saturates in controlling observed surprise, since the algorithm truncates low probability words to control surprise value (and the baseline surprise without any truncation is around five). Thus, to get better control over observed surprise values, we must truncate some more probable words as well, which would reduce the quality of the generated text, hence not considered here.

The observation on different growth rate of surprise values in top-$k$ and top-$p$ sampling in Fig. 2 is not very intuitive on its own. Our theoretical analysis in Sec. 3 helps explain nonlinear growth in cross-entropy rate in top-$k$ sampling and essentially linear growth in cross-entropy rate in top-$p$ sampling. Note that our theoretical analysis in Sec. 3 deals with cross-entropy while our experiments deal with cross-entropy rate. However, for practical purposes cross-entropy helps us give an intuition about cross-entropy rate in different sampling methods under stationary ergodic assumption. There is not much fluctuation in cross-entropy rate in Fig. 2c because we use feedback to control the cross-entropy rate more accurately, which gives accurate results even for a small number of tokens.

### 5.2 PERPLEXITY AND REPETITIONS

Here, we present some experimental observations for percentage of repeated tokens across different sampling methods and LMs. In Fig. 3, we generate texts with 200 tokens using different sampling methods and models with varying relevant input parameters such as $k$, $p$, or target surprise values, $\tau$.

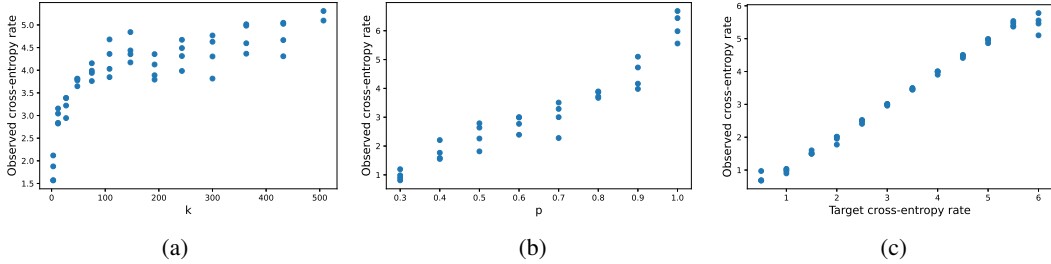

(a)             (b)             (c)

Figure 2: Cross-entropy rate in different sampling methods: (a) top-$k$, (b) top-$p$, and (c) mirostat. Note that cross-entropy rate is nonlinear in top-$k$ sampling and nearly linear in top-$p$ sampling. Also note that for a fixed input parameter in top-$k$ and top-$p$ sampling, observed cross-entropy rate fluctuates much, whereas mirostat shows negligible fluctuation in observed cross-entropy rate.

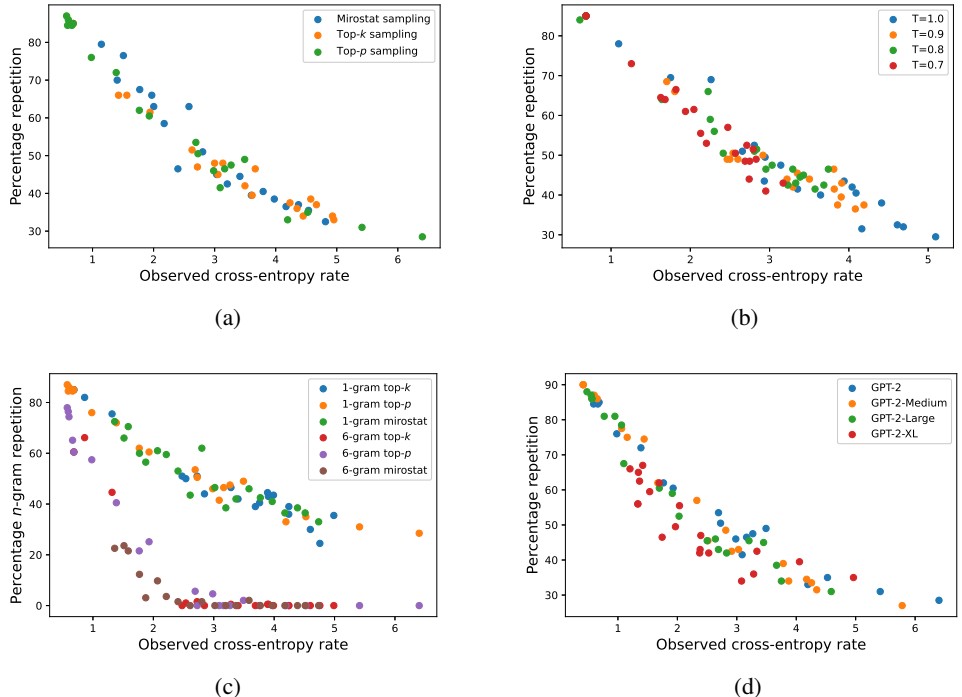

Figure 3: Percentage repetition vs. observed cross-entropy rate for (a) different sampling methods, (b) different temperature values, $T$, (c) for $n$-gram tokens and different sampling methods, (d) different LMs. Note that repetitions vary heavily with observed cross-entropy rate in the generated text and mostly independent of sampling method used. Further, large LMs like GPT-2-XL with 1558M parameters seem to have slightly less repetitions for the same cross-entropy rate.

We also consider the percentage of $n$-gram repetitions for different values of $n$ for a fixed sampling method. We define percentage $n$-gram repetition as $\left(1 - \frac{\text{number of distinct } n\text{-gram tokens}}{\text{total number of } n\text{-gram tokens}}\right) \times 100$, where an $n$-gram token simply means concatenation of $n$ contiguous tokens. Hence, for $n = 1$, $n$-gram repetitions capture word-level repetitions, whereas larger values of $n$ capture sentence-level repetitions. For $n = 1$, we refer to percentage 1-gram repetition simply as percentage repetition.

In Fig. 3a, we observe that percentage repetition decreases with increase in cross-entropy and more importantly, for a fixed GPT-2 model, this relation is independent of the sampling method.

In Fig. 3b, we use top-$k$ sampling with varying temperature values. We observe that repetitions for different temperature values and $k$ follow the same curve as in Fig. 3a. This implies cross-entropy

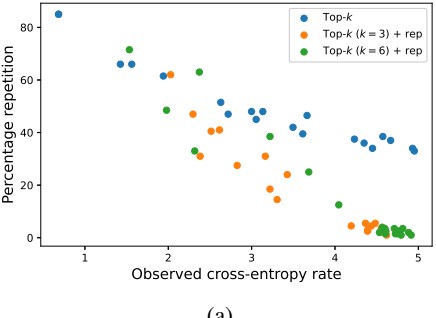 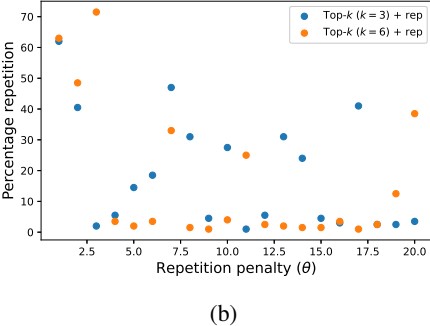

|  (a)  |  (b)  |

Figure 4: Percentage repetition vs. (a) observed cross-entropy rate for top-$k$ sampling with varying $k$, top-$k$ sampling with $k = 3, 6$ and varying repetition penalties, (b) repetition penalties for top-$k$ sampling with $k = 3, 6$.

controls the percentage repetitions in generated texts. Moreover, it implies that once the model and cross-entropy are fixed, percentage repetition is not affected by the considered sampling methods.

In Fig. 3c, we capture $n$-gram repetitions for varying cross-entropy rate and different values of $n$. We note from Fig. 3c that for small values of $n$, the percentage $n$-gram repetitions drop almost linearly with increase in cross-entropy, whereas for larger values of $n$, the percentage $n$-gram repetitions is very close to zero for cross-entropy greater than 3. This indicates sentence-level repetitions disappear after a threshold of cross-entropy whereas word-level repetitions continue to appear for larger values of cross-entropy. Also, note that in human-generated text data, there are often common pronouns and conjunctions that are essential and are often repeated, hence we do not expect a good sampling algorithm to have absolutely zero 1-gram repetitions. But, we do expect a good sampling algorithm to have minimum sentence-level repetitions, which all the sampling seems to show beyond a threshold of cross-entropy, which seems to be around 2.5 for GPT-2.

Fig. 3d plots percent repetition versus cross-entropy for different LMs using top-$p$ sampling for varying values of $p$. Larger LMs such as GPT-2-XL with 1558M parameters have slightly less repetitions for a fixed value of cross-entropy than smaller LMs such as GPT-2 with 117M parameters.

We also provide more experimental results showing near-linear relation between observed cross-entropy and percentage repetition in CTRL (Keskar et al., 2019) in Appendix F.

**Controlling repetitions using repetition penalty**   Consider a repetition penalty in top-$k$ sampling to reduce percent repetition, where we multiply negative scores of each repeated word in the vocabulary of GPT-2 by $\theta \in \{1, \ldots, 20\}$ before computing the corresponding probabilities using softmax. See Keskar et al. (2019) for more details on repetition penalty. Essentially, this reduces the probability of repeated words. In Fig. 4a, we observe that repetition penalty tends to reduce percent repetition for fixed cross-entropy rates. However, we also note in Fig. 4b that percent repetition does not have a good relation with $\theta$, which makes it difficult to use it in practice with target percent repetition. Note that the cases where percent repetition is close to zero have high-cross entropy rate, which is true for all values of $\theta$. Hence, we conclude that mirostat when used in conjunction with repetition penalty can provide high-quality results. However, we leave this investigation for future work. From these observations, we conclude that to control percentage repetition in generation we must control the cross-entropy of the output text, exactly like in mirostat.

## 5.3 BOREDOM AND CONFUSION TRAPS

Here we show top-$k$ and top-$p$ sampling cannot control output and therefore get trapped into low-quality generation, for a wide range of $k$ and $p$. We generated 10 samples of 900-token texts on the same context and averaged their observed cross-entropy at various points of generation in Fig. 5, except for the single-sample human-generated text (the tokens following the context in the corpus).

Fig. 5a illustrates that for small values of $k$ and $p$, both top-$k$ and top-$p$ sampling methods fall into low cross-entropy regions—boredom traps—which results in increase in repetitions as the length of

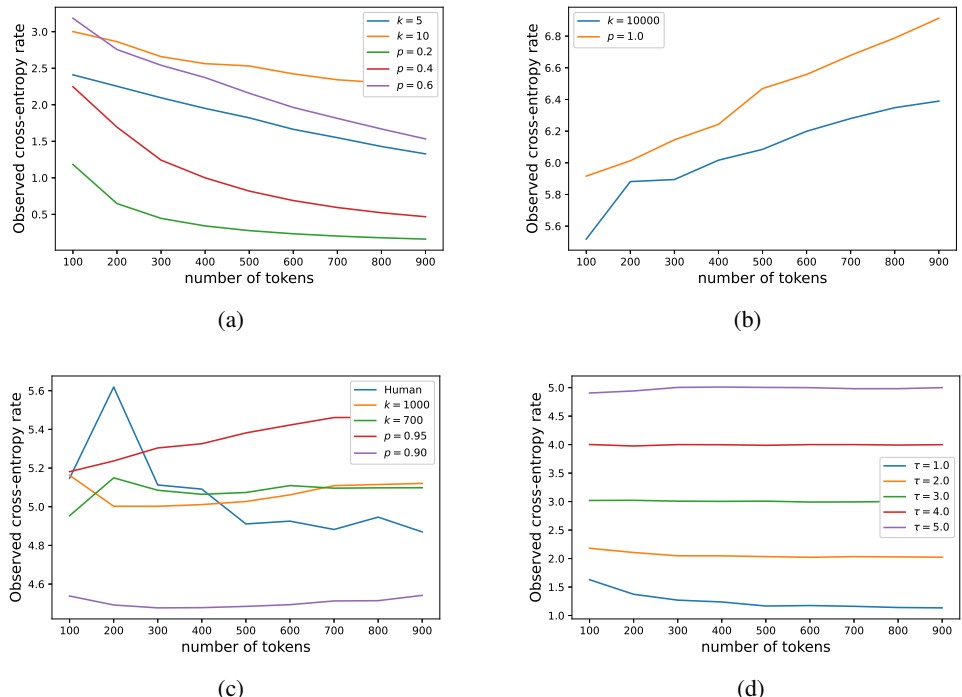

Figure 5: Cross-entropy rate vs. number of tokens generated for different sampling methods. We observe (a) boredom trap for small values of $k$ and $p$, (b) confusion trap for large values of $k$ and $p$, (c) human-like cross-entropy rate for moderate $k$ and $p$, (d) mirostat sampling that shows control over cross-entropy over varying lengths of texts.

the generated text increases, as illustrated in Sec. 5.2. Hence, lack of control over output statistics in these methods leads to degradation of quality in generated texts for longer texts.

Fig. 5b shows that for high values of $k$ and $p$ in top-$k$ and top-$p$ sampling methods respectively, the observed cross-entropy of the generated texts increases with the length of generated texts. This leads to increase in incoherence in the text as the token index increases—the confusion trap.

In Fig. 5c we choose certain values of $k$ and $p$ in an ad hoc manner and generate texts using top-$k$ and top-$p$ sampling methods respectively to observe that for these values of $k$ and $p$, the generated texts tend to have cross-entropy that seems to converge to a limiting value with increase in text length and not fall into either boredom or confusion traps. We also show how the observed cross-entropy varies with increase in text length in the human-generated text corresponding to the tokens following the context for these experiments. Human-generated text converges to some limiting value of cross-entropy when the generated text is long enough and does not fall into either boredom or confusion.

Finally, Fig. 5d shows cross-entropy for the mirostat-generated texts converges to the target cross-entropy within a few tokens and maintains the desired value of cross-entropy for long texts.

## 5.4 HUMAN EVALUATIONS

We evaluated performance using human raters, which further indicated the importance and necessity of controlling cross-entropy rates for generating high-quality texts. We generated 300 tokens using GPT-2 from a fixed context with average cross-entropy rate $\tau \in \{2.5, 3, 4, 5\}$ using both mirostat and top-$p$ sampling. We presented these texts and a human-generated 300 word continuation of the context to 43 participants from the University of Illinois at Urbana-Champaign and Indian Institute of Technology, Kanpur. Participants were not informed of the generation process and rated each text on 1 to 7 Likert scales for fluency, coherence, and overall quality. Further, the raters guessed if the text was AI- or human-generated. More details of the experiment are in Appendix D. Fig. 6 shows texts that had cross-entropy rate $\tau = 3$ received the best ratings by human participants for fluency,

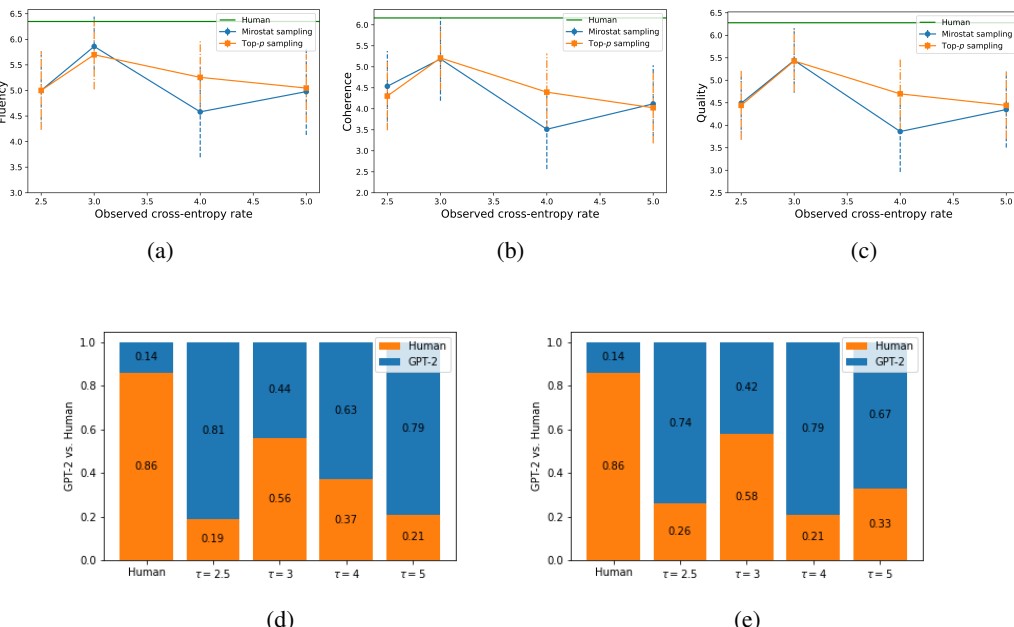

Figure 6: Human evaluation of top-$p$ and mirostat sampling for (a) fluency, (b) coherence, and (c) quality. Detecting human generated text from texts generated by GPT-2 under (d) top-$p$ sampling, (e) mirostat sampling. For a given observed cross-entropy rate, notice the similarity between human evaluations of top-$p$ and mirostat sampling across all values of observed cross-entropy rate.

coherence, and overall quality. Further, for $\tau = 3$, more than half of raters mistakenly guessed the AI-generated text to be human generated. These results show that controlling the cross-entropy rate helps generate high-quality human-like texts. Further, the sensitivity of these human evaluations to change in cross-entropy rates in Fig. 6 and the fluctuations in cross-entropy rates for fixed input parameters in top-$p$ and top-$k$ sampling from Fig. 2 show that mirostat produces high-quality texts without much ad hoc tuning of input parameters.

## 6 CONCLUSION

We provided a theoretical explanation of how perplexity varies as a function of input parameters in popular top-$k$ and top-$p$ neural text decoding algorithms, showing that log of perplexity varies nearly linearly as a function of $p$ and a highly nonlinearly as a function of $k$. Building on this analysis, we developed mirostat, a neural text decoding algorithm that directly controls the perplexity of the generated text over a wide range of text length. Notably, for longer texts and certain ranges of input parameters, top-$k$ and top-$p$ sampling fall into boredom and confusion traps which cause low-quality texts; Mirostat avoids both traps. Further, recent large-scale human evaluation of neural generated text suggests that human-judged text quality is maximized for a certain range of perplexity of the output: mirostat provides direct control to stay in that perplexity range. There are also implications for data compression as given in Appendix A.2. As a takeaway, we find that mirostat with target surprise around $3.0$, produces varying lengths of high-quality texts with minimal repetitions. This is corroborated in our own experiments with human raters.

We further analyze the relation between perplexity and repetitions in text: for fixed model, repetitions vary linearly with perplexity and are independent of the sampling method. We also find that larger LMs have less repetitions for any fixed perplexity. Future work would include theoretical analysis of repetitions, boredom and confusion traps, and convergence properties of mirostat.

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

## A  EXAMPLES

Here we describe two examples. The first example demonstrates the relation between surprise, repetitions, and incoherence. The second example demonstrates the relation between text compression and cross-entropy.

### A.1  RELATION BETWEEN SURPRISE, REPETITIONS, AND INCOHERENCE

**Example 1.** *We generate four samples of texts with 200 tokens (which are words or subwords that are part of the vocabulary of the model) using GPT-2 model with 117M parameters Radford et al. (2019). For the first two samples we use top-$p$ sampling with $p = 0.4$, for the third sample we use top-$p$ sampling with $p = 1.0$, and for the fourth sample we use our proposed mirostat sampling algorithm. The fifth example is human-generated, taken from the same corpus as the context. The context used is as follows.*

**Context:** *"For two months early in 1943, Shannon came into contact with the leading British mathematician Alan Turing. Turing had been posted to Washington to share with the U.S. Navy's cryptanalytic service the methods used by the British Government Code and Cypher School at Bletchley*

*Park to break the ciphers used by the Kriegsmarine U-boats in the north Atlantic Ocean. He was also interested in the encipherment of speech and to this end spent time at Bell Labs."*

1. **Observed average surprise value = 1.471     p = 0.4**

   **Generated text:** *"Turing's work on the cryptography of the Kriegsmarine was based on the cipher used by the Kriegsmarine's famous E-2. The Kriegsmarine code was based on the Cryptographic Algorithm (CAT) of the Royal Navy. It was a new method of solving the problem of encryption, which was a mathematical problem which was the basis of the cryptanalysis of the Japanese naval cipher.*

   *Turing's work on the cipher was based on the algorithm used by the Kriegsmarine's famous E-2. The cipher was based on the algorithm used by the Kriegsmarine's famous E-2.*

   *Turing's work on the cipher was based on the algorithm used by the Kriegsmarine's famous E-2.*

   *Turing's work on the cipher was based on the algorithm used by the Kriegsmarine's famous E-2.*

   *Turing's work on the cipher was based on"*

2. **Observed average surprise value = 1.887,     p = 0.4**

   **Generated text:** *"Turing was a great mathematician, and he was not a professional cryptographer. But he was a very talented cryptographer. He was also a highly intelligent man. He was a great man who was very well-versed in the law of probability. He was also very well-versed in the mathematical methods of the United States Army. He was also very well-versed in the mathematics of the Russian Academy of Sciences. He was also very well-versed in the mathematical methods of the German Academy of Sciences.*

   *In 1943, Shannon traveled to Moscow to meet with the Director of the NSA. He was told that the FBI had obtained information from the NSA about the German secret police. He was told that the FBI had obtained information from the NSA about the German secret police. He was told that the FBI had obtained information about the German secret police.*

   *The NSA, on the other hand, had been unable to get information from the NSA about the German secrete"*

3. **Observed average surprise value = 6.342     p = 1.0**

   **Generated text:** *"In January of 1942, in an open letter delivered to 8 persons and addressed to Sherwood, Shannon took the opportunity to address skeptics about the little out-of-print $128,000 unreported race as "cryptanalysis apparatus of extraterrestrial intelligence." His reply was also reprinted in 6 South Carolina publications (Atlanta Press, Fall Publishing, Miami, New House Press, Yorkshire Press, Sydney Press, Weatherby Press, Winter Press, and Thames & Hudson, to name a few), and eventually accepted as the final subscribed FHAT article. (Summary, were whom)*

   *Birmingham City Police concludes that [Turing's][new replace Robert] Symington is not guilty of breach of OKF law. Further, his suggestions for preventing his sentencing Should he reside anywhere near the city he defies extradition to Bavaria closer to dawn minimal? Lylimud Simpson (Ladies & Gentlemen, Lawrence Livermore University Press, Philadelphia): Jim Gilmore and its wife, Eva Civilsky,"*

4. **Observed average surprise value = 1.828     mirostat target surprise = 1.8**

   **Generated text:** *"In April 1943, Shannon was assigned to the U.S. Naval Intelligence Center at Naval Air Station Norfolk, Virginia, where he was assigned to work on a project to develop a new cipher for the U.S. Navy's cryptanalysis program. The project was to be conducted by the U.S. Navy's cryptanalysis team.*

   *In early April, the U.S. Navy was informed that Shannon had been assigned to work on an encrypted cipher for the Kriegsmarine's U-boat. The project was to be conducted by a group of cryptographers from the U.S. Naval Intelligence Center. The project was to be conducted by a group of cryptographers from the U.S. Naval Intelligence Center in a project that could be executed by the cryptanalysts.*

*In early April, Shannon was ordered by the U.S. Navy to provide the cryptanalysts with information on a cipher for the U-boat's U-boats that could"*

5. ***Observed average surprise value = 5.285        human-generated***

   ***Generated text:*** *"Shannon and Turing met at teatime in the cafeteria. Turing showed Shannon his 1936 paper that defined what is now known as the 'Universal Turing machine'. This impressed Shannon, as many of its ideas complemented his own.*

   *In 1945, as the war was coming to an end, the NDRC was issuing a summary of technical reports as a last step prior to its eventual closing down. Inside the volume on fire control, a special essay titled Data Smoothing and Prediction in Fire-Control Systems, coauthored by Shannon, Ralph Beebe Blackman, and Hendrik Wade Bode, formally treated the problem of smoothing the data in fire-control by analogy with 'the problem of separating a signal from interfering noise in communications systems'. In other words, it modeled the problem in terms of data and signal processing and thus heralded the coming of the Information Age. Shannon's work on cryptography was even more closely related to his later publications on communication theory. At the close of the war"*

Figure 7 shows plots of surprise values against indices of tokens in each of the samples in Ex. 1. The blue plot corresponds to surprise values of each token, while the red plot corresponds to average surprise values over a window of size 10 at each token index. Note that the surprise values drop drastically in Fig. 7a as the repetitions increase in Ex. 7.1. Similarly, in Fig. 7b, we observe a dip in surprise values wherever there is a repetition in Ex. 7.2. Clearly, there is a correlation between small average surprise values and repetitions. Further, in Fig. 7a note that the generating model seems to get trapped into a small surprise repetition region. We call this region of small surprise as *boredom trap*. We observe that these models tend to fall into a boredom trap for small values of $p$. Figure 7c corresponds to Ex. 7.3, where we choose $p = 1.0$ and illustrate that for large values of $p$, the average surprise value of the generated text tends to increase with the number of generated tokens, which leads to incoherence. We call this region of large surprise a *confusion trap*. Figure 7d shows surprise values corresponding to Ex. 7.4 which is generated using our proposed sampling algorithm, mirostat. We observe in Fig. 7d that mirostat increases the surprise value when when falling into a boredom trap and, thereby maintaining the average surprise value. By doing so, it not only helps generate high-quality text with predetermined average surprise value, but also helps avoid small surprise repetition regions and large surprise incoherent regions. In Fig. 7e, we show the surprise values in human-generated text that followed this context as shown in Ex. 7.5. We observe that human-generated text has average surprise value that is between values using top-$p$ sampling for $p = 0.4$ and $p = 1.0$. More importantly, human-generated text does not fall into either of the traps.

## A.2    TEXT GENERATION AND COMPRESSION

Here we will look at texts generated for various target surprise values using mirostat sampling with GPT-2 with 117M. We also observe the well-known relation between cross-entropy and data compression (Cover & Thomas, 2006, Ch. 5), (Gilbert, 1971). In particular, it is known that when the actual probability distribution of the generated text is not known, then the minimum lossless compression rate achievable is equal to the cross-entropy of the assumed distribution, which is the LM here, with respect to the actual unknown distribution, which is obtained from adaptive top-$k$ sampling here.

**Example 2.** *We generate 200 tokens for different values of target surprise values using the GPT-2 model with 117M parameters to show the quality of the text generated using Alg. 1 for different target surprise values. We also measure the compression rates obtained using arithmetic coding (Witten et al., 1987; Rissanen & Langdon, 1979) with the LM as the probability distribution. So, in a way, mirostat can generate text that has a predetermined minimum lossless compression rate for a given model.*

**Context:** *"For two months early in 1943, Shannon came into contact with the leading British mathematician Alan Turing. Turing had been posted to Washington to share with the U.S. Navy's cryptanalytic service the methods used by the British Government Code and Cypher School at Bletchley Park to break the ciphers used by the Kriegsmarine U-boats in the north Atlantic*

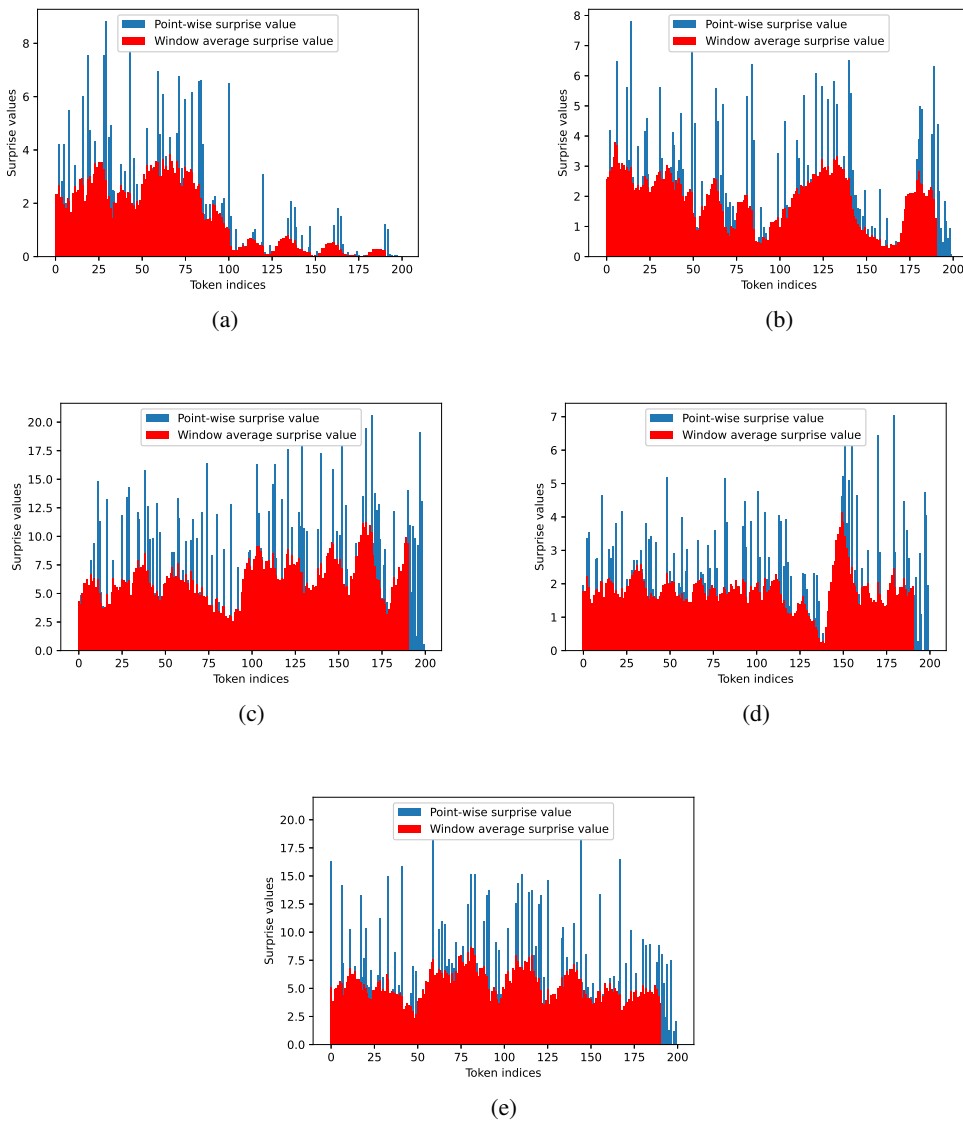

Figure 7: Relation between surprise values and attributes of generated text such as repetitions and incoherence: (a) top-$p$ sampling with $p = 0.4$ and average observed surprise = $1.471$, (b) top-$p$ sampling with $p = 0.4$ and average surprise = $1.887$, (c) top-$p$ sampling with $p = 1.0$ and average observed surprise = $6.342$, (d) mirostat sampling with target surprise, $\tau = 1.8$ and average observed surprise = $1.828$, (e) human-generated text, average observed surprise = $5.301$. We observe that repetitions in generated texts are correlated with dips in surprise values, whereas incoherence is correlated with large and increasing surprise values with indices.

*Ocean. He was also interested in the encipherment of speech and to this end spent time at Bell Labs."*

**Target surprise value = 1.0     Observed cross-entropy rate = 0.759     Average compressed size (per token) = 0.863     Percentage compression = 97.66%**

**Generated text:** *"In April 1943, Shannon was assigned to the U.S. Navy's cryptanalytic service, the Naval Intelligence Service. He was assigned to the cryptanalytic service's cryptanalysis unit, the Naval Intelligence Laboratory. He was also assigned to the cryptanalytic unit's cryptanalysis unit, the Naval Intelligence Laboratory.*

*In May 1943, Shannon was assigned to the cryptanalytic unit's cryptanalysis unit, the Naval Intelligence Laboratory. He was also assigned to the cryptanalytic unit's cryptanalysis unit, the Naval Intelligence Laboratory.*

*In June 1943, Shannon was assigned to the cryptanalytic unit's cryptanalysis unit. He was assigned to the cryptanalytic unit's cryptanalysis unit, the Naval Intelligence Laboratory.*

*In July 1943, Shannon was assigned to the cryptanalytic unit's cryptanalytic unit. He was assigned to the cryptanalytic unit's cryptanalytic unit, the Naval Intelligence Laboratory.*

*In August 1943, Shannon was assigned"*

**Target surprise value = 2.0     Observed cross-entropy rate = 1.987     Average compressed size (per token) = 2.07     Percentage compression = 95.13%**

**Generated text:** *"He was a member of the British Intelligence Service, and was also a member of the British Intelligence Service's Intelligence Advisory Board.*

*In early 1944, Shannon was assigned to the British Intelligence Service's Intelligence Advisory Board. He was also assigned the task of deciphering the ciphertext of the British Government Code. He was also assigned to the British Intelligence Service's Cryptanalytic Service. He was assigned to the British Intelligence Service's Intelligence Advisory Board at the end of 1944. He was assigned to the British Intelligence Service's Secret Intelligence Service at the end of 1944.*

*He was confirmed to work as the Director of the British Intelligence Service's Information Service at the end of 1944. He was the first British employee to be appointed as an Intelligence Advisory Officer at the end of the war. He was also confirmed as an Intelligence Advisory Officer to the United Kingdom's Intelligence Advisory Committee at the end of the war. (The United Kingdom is a member of the European Union.)*

*He was also appointed"*

**Target surprise value = 3.0     Observed cross-entropy rate = 3.002     Average compressed size (per token) = 3.141     Percentage compression = 91.84%**

**Generated text:** *"After a short stay at Bell Labs, Shannon was commissioned to report back to the Naval Intelligence Services (NISC) to learn more about his research. He made his way over to the Royal Naval Institute. He had already been working with a number of other cryptanalytic researchers, including the mathematician Alan Turing, for over eight years at Bell Labs, but this research was not yet done. He spent the next two months studying the cipher schemes of the various British cryptanalytic systems. The first step was to find out the actual cipher scheme for the cipher used by the HMS Queen Elizabeth on a submarine.*

*By the time Shannon arrived at the Royal Naval Institute he had been working for Patagonia Scientific and Technology, the British cryptanalytic company, for eight years. Patagonia received a call at 9 A.D. on the evening of September 4, 1943, asking to check the cipher system for a cipher used by the HMS Queen Elizabeth, which is written as:"*

**Target surprise value = 4.0     Observed cross-entropy rate = 3.943     Average compressed size (per token) = 4.06     Percentage compression = 89.98%**

**Generated text:** *"He also began writing code to help with deciphering encrypted communications. During 1943, Shannon kept a diary to help decipher his instructions to the cryptanalytic service. Hearing of the story of Alan Turing sent Shannon excitedly into a panic of his own. He that he had*

*only discovered Turing had made it clear to him that the great deception he had witnessed was due to the German government's efforts to conceal it. I believe he intended to reveal it. In the early days, the NSA was working with the National Security Agency (NSA) at different times against the Soviet Union in the early 90s and as recently as three years ago against the USSR in the early 1990s. It is believed that the four Russian spies were at the CIA while also an intermediary in the process of doing a disinformation campaign to manipulate the American public to support Bush Sr. by calling for American military intervention in Afghanistan against the Bush, Bush, and Cheney (and the Bush family) and by trying to persuade the public to"*

***Target surprise value = 5.0     Observed cross-entropy rate = 4.929     Average compressed size (per token) = 5.116     Percentage compression = 85.41%***
***Generated text:*** *"Naval Contractors hired Shannon as a shortage specialist at the Navy. So she worked alone with Turing to decipher the British cipher for the northern German submarines. Shannon undertook the work initially on the S.G.F. dual cipher. Shannon spent the whole working days at the Bell Labs lab at Bletchley Park.*

*After weeks of interrogations Shannon was able to break the great El-Fran jujito with a German accent. Shannon then calculated that this hydrocephalic consciousness alone would allow her to think the same words but without the huge amount of writing required to produce such a thing, and with millions of hours on board she was able to write some of the same phrases over a twenty-eight hours workweek.*

*Output: Individualist Hypothesis.*

*Miranda (alias Charlotte Reagmire, aka "The Lady in the Lake") (1945-2049) (wife of B.G. Lloyd) (Philadelphia, PA)"*

***Target surprise value = 6.0     Observed cross-entropy rate = 5.735     Average compressed size (per token) = 5.834     Percentage compression = 85.55%***
***Generated text:*** *"The CIA trained a small band of cryptanalysts to do the maths again, this time using a UK number generator. A few days after the wars Belgium introduced Bermuda rugby as the appropriate sport for the National Guard. Notwithstanding there being no convention around using English for Rugby at the time, there would be no possible instance in the history of fencing in Europe. Flags for the Hurricanes had evolved recently using a Dutch Italian design called the Crazy Flag. These flag designs come largely of British origin and the date published of its introduction by the Royal Armouries of Lameucers is from 1638. The camouflage was recently added to the new Irish power orange flag. The design is based on the weapons bao mouèret Standard and has two coloured pouches connected to the rifle barrel by two checks along the top of the barrel with protection straps around the barrel to protect the cutouts. NATO hired a team of physicists to do the reconstruction. Readers who want to know more about this new"*

In Ex. 2 we can see that low value of surprise value results in repetitions and high value of surprise value results in incoherent generated texts. Moderate surprise values result in good quality, coherent text with no repetition. Also, note that the control does not work well when the target surprise value is greater then 5. This is because without any truncation, the average surprise of pure sampled text comes out to be around 5.4. Thus, in order to attain higher values of average surprise, we need to truncate from both sides of the distribution.

## B  THEORETICAL RESULTS

**Theorem 1.** *If words are sampled from the Zipf's distribution given by (1), then the surprise value of a word with rank $k$ and its rate of increase are given by*

$$\mathfrak{S}(k) = s \log k + \log H_{N,s}, \tag{3}$$

$$\frac{d\mathfrak{S}(x)}{dx} = \frac{s}{x} \tag{4}$$

*respectively, where $\mathfrak{S}(x)$ is a continuous function with the same expression as $\mathfrak{S}(k)$ with a continuous domain.*

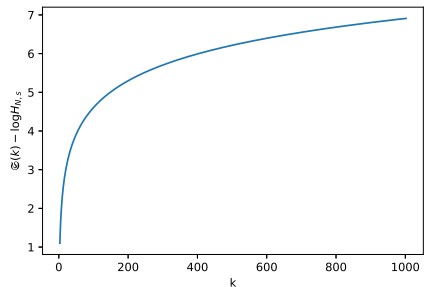 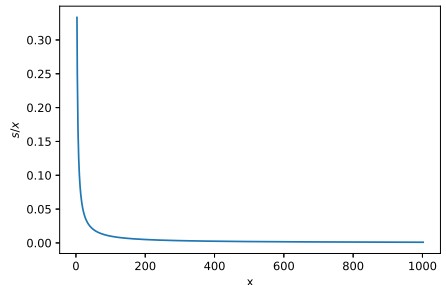

(a) Plot of $\mathfrak{S}(k) - \log H_{N,s}$ vs. $k$ for $s = 1$, $N = 50,000$.

(b) Plot of $s/x$ vs. $x$ for $s = 1$, $N = 50,000$.

Figure 8: Theoretical analysis of surprise values in top-$k$ sampling under Zipfian statistics. We note that surprise values increase sharply for small values of $k$, whereas surprise values hardly change for large values of $k$.

*Proof.* The expression of $\mathfrak{S}(k)$ follows directly from Def. 1 and (1). $\qquad\square$

From Fig. 8, we note that $\mathfrak{S}(x)$ is highly sensitive to change in $x$ for small values of $x$ and its sensitivity to $x$ decreases drastically with increase in $x$. Now, we analyze how cross-entropy varies with $k$. Let $P_M$ be the model distribution. Top-$k$ sampling works by truncating the tail of the distribution $P_M$ and samples from the most probable $k$ tokens. Let the truncated distribution be denoted by $P_{M_k}$. In Prop. 1, we provide an expression for $H(P_{M_k}, P_M)$.

**Proposition 1.** *Let $P_M$ be the model distribution satisfying (1) with vocabulary of size $N$ and let $P_{M_k}$ be the model distribution obtained by top-$k$ sampling. Then $H(P_{M_k}, P_M)$ is given by*

$$H(P_{M_k}, P_M) = \frac{s}{H_{k,s}} \sum_{i=1}^{k} \frac{\log i}{i^s} + \log H_{N,s}. \tag{5}$$

*Proof.* The distribution $P_M$ is given by (1) with vocabulary size $N$, and it is easy to check that the distribution $P_{M_k}$ corresponding to top-$k$ sampling is also given by (1) but with vocabulary size $k$. The rest follows directly from the definition of cross-entropy in Sec. 2. $\qquad\square$

It is difficult to get an intuition about the behavior of $H(P_{M_k}, P_M)$ directly from (5). Thus, in Thm. 2 we obtain an approximation to $H(P_{M_k}, P_M)$ that shows $H(P_{M_k}, P_M)$ is essentially of the form $c_1(1 - c_2 \frac{\ln k + c_3}{k^\epsilon - 1}) + c_4$ for $0 < \epsilon < \frac{1}{\ln 2}$, where $c_1, c_2, c_3, c_4$ are some constants. Hence we observe that $H(P_{M_k}, P_M)$ grows fast with small values of $k$ and slows down for large values of $k$.

**Theorem 2.** *Let $P_M$ be the model distribution satisfying (1) with vocabulary of size $N$. and let $P_{M_k}$ be the model distribution obtained using top-$k$ sampling. Then, for $1 < s \leq \frac{1}{\ln 2}$, $H(P_{M_k}, P_M)$ can be approximated as*

$$H(P_{M_k}, P_M) \approx \frac{b_1 \epsilon}{b_3} \left(1 - \frac{b_2 b_3 (\ln k + \frac{1}{\epsilon}) - b_1}{b_1(b_3 k^\epsilon - 1)}\right) + \log H_{N,s}, \tag{6}$$

*where $b_1 = s\left(\frac{\log 2}{2^{1+\epsilon}} + \frac{\log 3}{3^{1+\epsilon}} + \frac{1}{\epsilon(\ln 2)3^\epsilon}\left(\ln 3 + \frac{1}{\epsilon}\right)\right)$, $b_2 = \frac{s}{\epsilon \ln 2}$, and $b_3 = 1 + 0.7\epsilon$ are constants.*

*Proof.* From Prop. 1 we have $H(P_{M_k}, P_M) = \frac{s}{H_{k,s}} \sum_{i=1}^{k} \frac{\log i}{i^s} + \log H_{N,s}$. We start by finding bounds for the expression $\sum_{i=1}^{k} \frac{\log i}{i^s}$.

First note that the function $\frac{\log t}{t^s}$ is a decreasing function of $t$ for $t > e^{\frac{1}{s}}$. Thus, for $1 \le s \le \frac{1}{\ln 2}$, we have the following inequalities

$$\frac{\log 2}{2^s} + \int_3^{k+1} \frac{\log t}{t^s} dt \le \sum_{i=1}^k \frac{\log i}{i^s} \le \frac{\log 2}{2^s} + \frac{\log 3}{3^s} + \int_4^{k+1} \frac{\log (t-1)}{(t-1)^s} dt. \tag{7}$$

Solving the above integration for $1 < s \le \frac{1}{\ln 2}$ we get

$$\frac{a_1}{H_{k,s}} - \frac{a_2}{H_{k,s}(k+1)^\epsilon} \left( \ln (k+1) + \frac{1}{\epsilon} \right) \le H(P_{M_k}, P_M) - \log H_{N,s} \le \frac{b_1}{H_{k,s}} - \frac{b_2}{H_{k,s}k^\epsilon} \left( \ln k + \frac{1}{\epsilon} \right), \tag{8}$$

where $a_1 = s \left( \frac{\log 2}{2^{1+\epsilon}} + \frac{1}{\epsilon(\ln 2)3^\epsilon} \left( \ln 3 + \frac{1}{\epsilon} \right) \right)$, $a_2 = \frac{s}{\epsilon \ln 2}$, $b_1 = s \left( \frac{\log 2}{2^{1+\epsilon}} + \frac{\log 3}{3^{1+\epsilon}} + \frac{1}{\epsilon(\ln 2)3^\epsilon} \left( \ln 3 + \frac{1}{\epsilon} \right) \right)$, $b_2 = \frac{s}{\epsilon \ln 2}$.

Now, we bound $H_{k,s}$ as follows. Note that $\frac{1}{t^s}$ is a decreasing function in $t$ for $t > 0$ and $s > 0$, hence, we have

$$\int_1^{k+1} \frac{1}{t^s} dt \le \sum_{i=1}^k \frac{1}{i^s} \le 1 + \int_2^{k+1} \frac{1}{(t-1)^s} dt \tag{9}$$

$$\frac{1 - (k+1)^{-\epsilon}}{\epsilon} \le \sum_{i=1}^k \frac{1}{i^s} \le 1 + \frac{1 - k^{-\epsilon}}{\epsilon}. \tag{10}$$

We empirically observed that $H_{k,s}$ can be approximated well as

$$H_{k,s} \approx 0.7 + \frac{1 - k^{-\epsilon}}{\epsilon}, \tag{11}$$

which lies between the bounds found in (10). Moreover, we approximate $H(P_{M_k}, P_M)$ using the upper bound obtained in (7) to get

$$H(P_{M_k}, P_M) \approx \frac{1}{H_{k,s}} \left( b_1 - \frac{b_2}{k^\epsilon} \left( \ln k + \frac{1}{\epsilon} \right) \right) + \log H_{N,s} \tag{12}$$

$$\approx \frac{\epsilon}{b_3(1 - \frac{k^{-\epsilon}}{b_3})} \left( b_1 - \frac{b_2}{k^\epsilon} \left( \ln k + \frac{1}{\epsilon} \right) \right) + \log H_{N,s} \tag{13}$$

$$\approx \frac{b_1 \epsilon}{b_3} \left( 1 - \frac{b_2 b_3 (\ln k + \frac{1}{\epsilon}) - b_1}{b_1 (b_3 k^\epsilon - 1)} \right) + \log H_{N,s}, \tag{14}$$

where (14) follows by writing $\frac{1}{(1 - \frac{k^{-\epsilon}}{b_3})}$ as an infinite series in (13), then simplifying the expression and writing the infinite series back as a fraction. $\square$

In Thm. 3, we provide approximate expressions for $\mathfrak{S}(p)$ and $\frac{d\mathfrak{S}(p)}{dp}$ that shows that $\mathfrak{S}(p)$ grows essentially linearly with $p$.

**Theorem 3.** *If words are sampled from the Zipf's distribution given by (1). If $\epsilon > 0$ is a small constant, then $\mathfrak{S}(p)$ and the rate of change of $\mathfrak{S}(p)$ with respect to $p$ is given by*

$$\mathfrak{S}(p) \approx \frac{(1+\epsilon)}{b \ln 2} H_{N,s} p - \frac{(1+\epsilon)}{\epsilon} \log b + \log H_{N,s} \tag{15}$$

$$\frac{d\mathfrak{S}(p)}{dp} \approx \frac{(1+\epsilon)H_{N,s}}{b \ln 2} (1 + \frac{H_{N,s}\epsilon p}{b}), \tag{16}$$

*where $b = 1 + 0.7\epsilon$.*

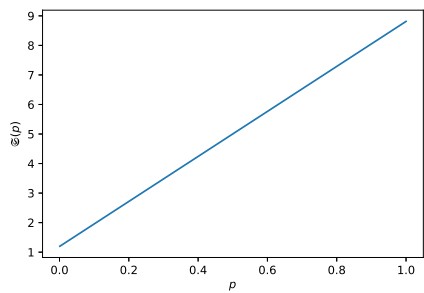
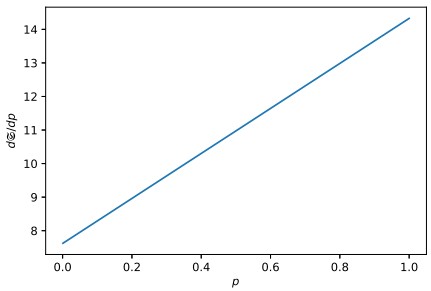

(a) Plot of $\mathfrak{S}(p)$ vs. $p$ for $s = 1.2$, $N = 50,000$.

(b) Plot of $\frac{d\mathfrak{S}(p)}{dp}$ vs. $p$ for $s = 1.2$, $N = 50,000$.

Figure 9: Theoretical analysis of surprise values in top-$p$ sampling. We observe that unlike top-$k$ sampling, surprise values grow linearly with $p$ in top-$p$ sampling.

*Proof.* The cumulative probability $p(k)$ for Zipf's distribution is given by $p(k) = \frac{H_{k,s}}{H_{N,s}}$. Using the approximation to $H_{k,s}$ in (11), we have

$$p(k) = \frac{b - k^{-\epsilon}}{\epsilon H_{N,s}}, \tag{17}$$

where $b = 1 + 0.7\epsilon$.

Now, writing $k$ as a function of $p$, we get

$$k = (b - \epsilon p H_{N,s})^{-\frac{1}{\epsilon}}. \tag{18}$$

Using (18) in the equation $\mathfrak{S}(x) = s \log x + \log H_{N,s}$ from Thm. 1, we get

$$\mathfrak{S}(p) = -\frac{1+\epsilon}{\epsilon} \log (b - H_{N,s}\epsilon p) + \log H_{N,s}$$

$$= -\frac{1+\epsilon}{\epsilon \ln 2} \ln (1 - \frac{H_{N,s}\epsilon p}{b}) - \frac{1+\epsilon}{\epsilon} \log b + \log H_{N,s}. \tag{19}$$

Further, taking $\epsilon$ small enough, we can approximate $\ln (1 - \frac{H_{N,s}\epsilon p}{b}) \approx -\frac{H_{N,s}\epsilon p}{b}$. Thus, we have

$$\mathfrak{S}(p) \approx \frac{(1+\epsilon)}{b \ln 2} H_{N,s} p - \frac{(1+\epsilon)}{\epsilon} \log b + \log H_{N,s}. \tag{20}$$

Now, $\frac{d\mathfrak{S}(p)}{dp}$ can be directly computed from (19) as

$$\frac{d\mathfrak{S}(p)}{dp} = \frac{H_{N,s}(1+\epsilon)}{\ln 2(b - H_{N,s}\epsilon p)}. \tag{21}$$

For $\epsilon$ small enough, we can use the approximation $\frac{1}{1 - \frac{H_{N,s}\epsilon p}{b}} \approx 1 + \frac{H_{N,s}\epsilon p}{b}$ which gives

$$\frac{d\mathfrak{S}(p)}{dp} = \frac{H_{N,s}(1+\epsilon)}{b \ln 2} \left(1 + \frac{H_{N,s}\epsilon p}{b}\right). \tag{22}$$

$\square$

In Fig. 9a, we plot the approximate expression for $\mathfrak{S}(p)$ obtained in Thm. 3 which is a linear function in $p$ and has a slope approximately 10 for $s = 1.07$ and $N = 50,000$. In Fig. 9b, we plot the approximate expression for $\frac{d\mathfrak{S}(p)}{dp}$ from Thm. 3 which is also a linear function of $p$. This tells us that even though $\mathfrak{S}(p)$ can be approximated as essentially a linear function of $p$, it has a slightly increasing slope. Further, unlike the plot of $\frac{d\mathfrak{S}(x)}{dx}$ in Fig. 8b, which is decreasing with $k$, $\frac{d\mathfrak{S}(p)}{dp}$ in Fig. 9b has a positive slope. Next, we provide an approximate expression for $H(P_{M_p}, P_M)$ showing that it grows near-linearly with $p$.

**Theorem 4.** *Let $P_M$ be the model distribution satisfying (1) with vocabulary of size $N$. and let $P_{M_{k(p)}}$ be the model distribution obtained using top-p sampling where $k(p)$ is the minimum value of $k$ satisfying $\frac{1}{H_{N,s}} \sum_{i=1}^{k(p)} \frac{1}{i^s} \geq p$. Then, for $1 < s \leq \frac{1}{\ln 2}$, $H(P_{M_p}, P_M)$ can be approximated as*

$$H(P_{M_p}, P_M) \approx \frac{s}{2 \ln 2} \left( p H_{N,s} + \epsilon p^2 H_{N,s}^2 \right) + \log H_{N,s}. \tag{23}$$

*Proof.* The cumulative probability $p(k)$ for (1) can be written as

$$p(k) = \frac{H_{k,s}}{H_{N,s}}. \tag{24}$$

We approximate $\sum_{i=1}^{k} \frac{\ln i}{i^s} \approx \int_1^k \frac{\ln t}{t^s} dt$ to get

$$H(P_{M_p}, P_M) \approx \frac{s}{p H_{N,s} \ln 2} \left( \int_1^k \frac{\ln t}{t^s} dt \right) + \log H_{N,s}, \tag{25}$$

$$= \frac{s}{p H_{N,s} \ln 2} \left( \frac{1}{\epsilon^2} - \frac{1}{\epsilon k^\epsilon} (\ln k + \frac{1}{\epsilon}) \right) + \log H_{N,s}, \tag{26}$$

$$\tag{27}$$

Approximating $p(k)$ from (24) as $p(k) = \frac{1}{H_{N,s}} \int_1^k \frac{1}{t^s} dt$, we get

$$k = (1 - \epsilon p H_{N,s})^{-\frac{1}{\epsilon}}. \tag{28}$$

Using (28) in (26), we have

$$H(P_{M_p}, P_M) \approx \frac{s}{p H_{N,s} \ln 2} \left( \frac{1}{\epsilon^2} - \frac{1}{\epsilon k^\epsilon} (\ln k + \frac{1}{\epsilon}) \right) + \log H_{N,s},$$

$$= \frac{s}{p H_{N,s} \ln 2} \left( \frac{1}{\epsilon^2} + \frac{(1 - \epsilon p H_{N,s})}{\epsilon^2} (\ln (1 - \epsilon p H_{N,s}) - 1) \right) + \log H_{N,s}$$

$$= \frac{s}{\epsilon^2 p H_{N,s} \ln 2} \left( 1 + (1 - \epsilon p H_{N,s})(\ln (1 - \epsilon p H_{N,s}) - 1) \right) + \log H_{N,s}$$

$$= \frac{s}{\epsilon^2 p H_{N,s} \ln 2} \left( \ln (1 - \epsilon p H_{N,s}) - \epsilon p H_{N,s} \ln (1 - \epsilon p H_{N,s}) + \epsilon p H_{N,s} \right) + \log H_{N,s}$$

$$\approx \frac{s}{2 \ln 2} \left( p H_{N,s} + \epsilon p^2 H_{N,s}^2 \right) + \log H_{N,s}, \tag{29}$$

where (29) is obtained by taking the approximation $\ln (1 - \epsilon p H_{N,s}) \approx -\epsilon p H_{N,s} - \frac{(\epsilon p H_{N,s})^2}{2}$ for sufficiently small $\epsilon p H_{N,s}$. $\qquad\square$

## C   MMSE ESTIMATION OF ZIPF'S EXPONENT

We assume words follow Zipf's distribution (1). Further, we observe the probabilities produced by our LM as $\{p^{obs}(1), \ldots, p^{obs}(i), \ldots, p^{obs}(N)\}$. We use minimum mean-squared error (MMSE) estimation to find the value of $s$. However, $s$ shows up in $p(k; s, N)$ both as an exponent of $k$ and in $H_{N,s}$ which makes the computation difficult. Hence we estimate by minimizing MSE between logarithm of ratios of subsequent probabilities which eliminates $H_{N,s}$, i.e. we estimate $s$ as

$$\hat{s} = \arg\min_s \sum_{i=1}^{N-1} (st_i - b_i)^2 = \frac{\sum_{i=1}^{N-1} t_i b_i}{\sum_{i=1}^{N-1} t_i^2}, \tag{30}$$

where $t_i = \log \frac{i+1}{i}$ and $b_i = \log \frac{p^{obs}(i)}{p^{obs}(i+1)}$. When $N$ is large, we estimate $s$ using the most probable $m$ tokens for $m$ around 100 to improve time complexity, which gives a practically good estimate.

## D  HUMAN EVALUATIONS: EXPERIMENTAL SETUP

For human evaluations, 43 participants were shown nine text samples each of 300 words generated by mirostat, human, and top-$p$ sampling shown in Tab. 1, Tab. 2, Tab. 3 in a random but fixed order. These texts were generated using the same context as in Ex. 1, which was also shown to the participants. For mirostat sampling, we simply set the required value of $\tau \in \{2.5, 3, 4, 5\}$ to obtain the sample text, but for top-$p$ sampling, we sampled several times with different values of $p$ till we obtained samples that had observed cross-entropy in $\{2.5, 3, 4, 5\}$ such that comparisons between top-$p$ and mirostat can be made. The participants were not shown the method of generation of these texts or any of its statistical properties. Each participant was asked to rate the fluency, coherence, and quality of the generated text on a scale of 1 to 7, where 1 is the worst possible rating and 7 is the best possible rating. Further, they were asked to guess whether the text was generated by an AI algorithm or a human. The participants were also provided with the standard definitions of fluency and coherence. They were asked to rate themselves on their knowledge of English on a scale of 1 to 5, where 1 meant no knowledge of English and 5 meant proficient in English and the participants rated their knowledge of English as 4.3 on an average.

## E  ALTERNATE ALGORITHMS TO CONTROL PERPLEXITY

Here we provide two alternative algorithms to Alg. 1 that also control perplexity and compare their performance.

**Mirostat 2.0**   Here we provide an alternate algorithm for perplexity control, Alg. 2, which does not depend on the distribution of the underlying LM. In this sense, Alg. 2 controls perplexity in more general sequential generative models than Alg. 1 where the underlying distribution may not be Zipfian. In our work, we choose Alg. 1 since it has only an additional constant time complexity compared to top-$k$ sampling. Whereas Alg. 2 has additional time complexity that depends on target cross-entropy rate and vocabulary size, which may vary with different LMs. Moreover, since we are working specifically with languages here, which have Zipfian properties (Zipf, 1965; Piantadosi, 2014; Lestrade, 2017) and since Alg. 1 empirically provides good control on perplexity of generated text, we choose this algorithm for our human experiments, which also validates the performance of Alg. 1.

---

**Algorithm 2:** Mirostat 2, an alternate implementation of mirostat sampling for perplexity control

---

Target cross-entropy $\tau$, maximum cross-entropy $\mu = 2\tau$, learning rate $\eta$
**while** *more words are to be generated* **do**
    Sort the words in descending order of their surprise values
    Truncate the words with surprise values greater than $\mu$
    Normalize the probabilities of the remaining words
    Sample the next word $X$ from the remaining words
    Compute error: $e = \mathfrak{S}(X) - \tau$
    Update $\mu$: $\mu = \mu - \eta e$
**end**

---

In Fig. 10, we compare Alg. 1 and Alg. 2 in terms of their control on cross-entropy rates and the relation between cross-entropy rate and percentage $n$-gram repetitions. We find that both the algorithms perform almost identically both in terms of controlling perplexity and ability to control repetition.

**Mirostat average**   Here, we look at Alg. 3 which is identical to Alg. 2 except that for computing the error term, we use the average surprise of generated text instead of using the surprise of the most recently generated word.

In Fig. 11, we find that Alg. 3 performs poorly compared to Alg. 1 both in terms of controlling perplexity and repetitions. This is interesting since, intuitively, Alg. 3 should control observed cross-entropy rate well since we compute error from the observed cross-entropy rate itself instead of the surprise value of the most recent word. Our understanding for poor performance of why Alg. 3 does not control cross-entropy rate well is that the surprise values of words change very abruptly in

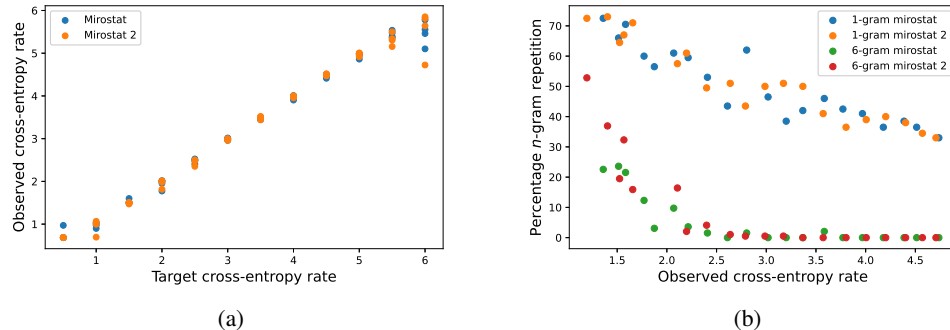

(a)                                    (b)

Figure 10: Comparing mirostat Alg. 1 with mirostat 2 Alg. 2: (a) Observed vs. target cross-entropy rate, (b) percentage $n$-gram repetitions vs. observed cross-entropy rate. We observe that both algorithms give equal performance in terms of controlling cross-entropy and repetitions.

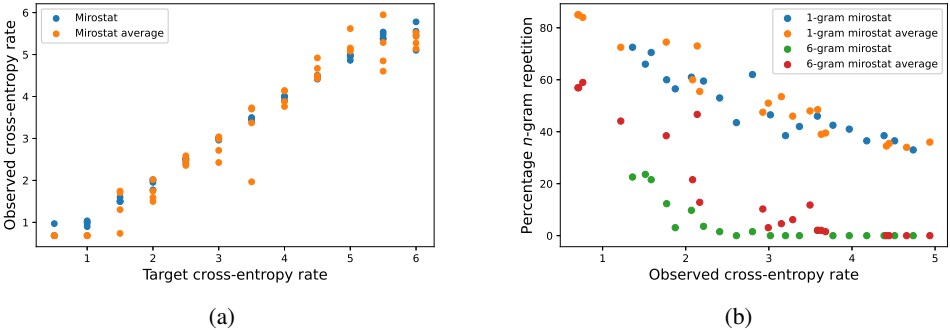

(a)                                    (b)

Figure 11: Comparing mirostat Alg. 1 with mirostat average Alg. 3: (a) Observed vs. target cross-entropy rate, (b) percentage $n$-gram repetitions vs. observed cross-entropy rate. We observe that mirostat average performs poorly in controlling cross-entropy and has worse 6-gram percentage repetition vs. observed cross-entropy rate curve.

| Method | Observed cross-entropy | Generated text |
|---|---|---|
| Mirostat, $\tau = 2.5$ | 2.5 | Shannon was also interested in the development of the computer program for the decoding of the German military radio signals.
In April 1944 Shannon was assigned to the U.S. Commandant's Office in Berlin. He was assigned to the U.S. Commandant's Office in Berlin as part of the Operations in Europe and Central Asia Command. He was also a member of the Commandant's Staff of the U.S. Army, with which he was familiar in the field of military technology.
In May 1944, Shannon was assigned with the U.S. Army Counterintelligence Center, located at Fort Irwin, California. During this time he met and worked with the U.S. Army Commandant Assistant to the President, the Chief of Staff of the U.S. Army General Staff and a young man with an interesting background in cryptography. In May 1944, Shannon received a call from one of the senior officers of the U.S. General Staff. "Mr. Shannon, I think we have some good news for you," said a senior officer at the Army Intelligence Headquarters in Quantico, Virginia, who was in charge of the establishment of the counterintelligence center at Fort Irwin. "From there, it went on to be established by the U.S. Military Intelligence Officers. It is a very good organization. It has been in operation since the end of the war."

Over the next several years Shannon was stationed at Fort Irwin as part of the Department of Defense |
| Mirostat, $\tau = 3$ | 3 | In April 1943, Shannon attended a conference on data security at the University of California, Berkeley, held on the topic of security issues. The event was organised by the Theoretical Computer Engineering Society, a group of computer scientists, statisticians, engineers and journalists who are interested in security issues, but who are not interested in privacy concerns or in privacy issues.
In the following month, Shannon helped to organise the meeting and the conference. He also attended the meeting of the National Security Council, and the meetings of the NSC and other federal agencies and agencies. In the early days of the war, using the NSC's computer systems Shannon used his knowledge of cryptography to create a series of cryptosystems that he had incorporated into his code, which included the AES cipher, which was the first of the AES-based cryptosystems.
It was on the 19th of January (January 1943) that Shannon would begin to use his knowledge of technology in his code and his code was published on the Internet.

On 1st of Oct 1943, Shannon was invited to the National Academy of Sciences' annual meeting. The NSC

hosted it in Washington, DC for the first time five days before the meeting. The ceremony of the meeting,

which included the signing of the Declaration on the Establishment of the National Academy of Sciences

and the signing of the American Educational System's Center for Information Technology and Society, and

the signing of its National Security Directive, was held |
| Mirostat, $\tau = 4$ | 4 | Indeed, it seemed to Shannon that MIT was the perfect place to learn the great secrets of mathematics, particularly about cryptography. And it soon became clear that she was spot on. Shannon described in detail how every single mathematical task in the cryptanalysis process was based on mathematics.
After years of re-enacting in a variety of settings, including short films and video games, Shannon began to embrace the science. She invented the Matrix, a program that combined the information from a computer simulation with real-world data.
"In October, I contacted myself and said, 'Do you think I'll be able to put this into Manhattan in a little bit, with an wireless connection?'" she said to a journalist. "I completely understood that the world outside of New York was amazing, and I was sort of reeling from what I thought the world outside of New York was doing. Indeed, that was precisely the bizarre thing, because I thought math was actually so much more complex than that, especially in mathematics. I was really mystified by everything: what I was investigating for my own personal reasons, as I've been doing to a lot of other people, and this was some amazingly complex thing that was just so complex, that was all the more fascinating."

After several months of researching and looking at mathematical techniques for creating the world around

us, Shannon, along with collaborators Chellie Rieder and John D. Riddell from Carnegie Mellon's Com-

puter Science Department |
| Mirostat, $\tau = 5$ | 5 | His work with Fort Bragg, New York, and Kritzenberg proved to be the key to the development of as much cyberspace in the space program as any German brain establishing a new language was known to have produced. But Tucker did a better job of penetrating the ciphers as his 'science' than did Turing. Nearly a decade later, he devoted another year to having his library prepared by him in the United States by the weavers (who, like Turing, so much liked mathematics). The book that made him such a personality (which was to be lavishly praised by the most, but also, by all, respected mathematicians of the North American world of public art and theatre and the cinema) was The Solid Code, a fine collection of lectures and correspondence between Shannon as well as some of the larger scale cryptanalysts in the U.S. They argued the foundations of the calculus, giving possession to finite numbers and a basic sequence to all possible operations, and they presented a new way of explaining the physics which could be analysed under conditions of finite energy and the presence of awareness. In other words, Turing was working in relations with the world.

While conducting C-level research on the atomic bomb, Lerner gave him an opportunity to examine a

previously unpublished study in Bletchley Park. It established the basic crystal structure of the nucleus of

a cubic centimetre of amber glass in which the atoms hold together indefinitely. The emitter was activated

by nucle |

Table 1: Mirostat generated text data for human evaluations

the generation process whereas the average value is rather smooth and does not give a good feedback of the current state of the generation process, hence, making it difficult to control the cross-entropy rate. However, we leave any theoretical result on the performance this algorithm to future work.

| Method | Observed cross-entropy | Generated text |
|---|---|---|
| Top-$p$, $p = 0.56$ | 2.5 | At the Bell Labs he developed the Advanced Encryption Standard (AES) for machine learning. Shannon's idea was to create a set of C code, each of which would be associated with a single letter. A ciphers bit was a C program that would be used to generate the ciphers. Shannon's approach was to use an algorithm called a ciphers hash to generate the ciphers. The first time Shannon had written a ciphers bit he did not know how to generate the ciphers. In the next couple of years he was developing a method to write a code that would be used to generate the ciphers. In January 1944 Shannon started working on a new cipher called a ternary cipher. He began working on a ternary cipher called an xor. The first ternary cipher was produced by Dr. H.G. Cooper, a German cryptographer who had worked on the first xor. It was later revealed that the ternary cipher was a work in progress. However, the ternary cipher was the first known cryptographically secure cipher.

The New York Times published a story in January 1945 that stated that "a British engineer working on a ternary cipher was having trouble understanding how to use a ternary cipher. Dr. A.G. Cooper, a cryptographer at Bell Labs, had been working on a ternary cipher since 1941. He had developed a |
| Top-$p$, $p = 0.65$ | 3 | He also came to the conclusion that there was a need for the invention of new methods for decrypting human speech. It was not until January 1944 that Shannon was able to confirm that a single piece of information was required to decrypt the code and the first known technique was the "Multi-Level Encryption" method. In this method a special algorithm was used to decrypt a message. This method, called the "Cypher program", was created by David Klein, a cryptographer and author of a book called "What Cryptographers Do: An Inside Look at the Cryptography of the United States Government". It was a technique that was first demonstrated in a presentation Shannon gave at the 1944 SIGGRAPH conference. The "Cypher program" was to be implemented in software as part of the "Interrogation" procedure of the White House Office of National Intelligence. The program was designed to be a cipher cipher that would decrypt the messages of the American government and was to be used to decrypt other governments' communications. The main function of the multi-level encryption method was to "decrypt" human speech. Shannon also created the White House Office of Information Security. The White House Office of Information Security developed the first non-state computer-readable language and introduced the information-processing unit of the National Security Agency.

In the summer of 1944, Shannon was assigned to investigate a cryptogram-encoded message. He was able to analyze it using his computer, and |
| Top-$p$, $p = 0.85$ | 4 | It was at Bell Labs that Shannon wrote his first book, In Plain Sight, a book for a year in 1961 called Non-verbal Behavior and a periodical for which Shannon was selected by James Clerk Maxwell as a reviewer. It was a radical assessment of behaviour and that was the basis of the book's first chapter, "My Love Letters." In 1961, Shannon wrote her bestseller The Shock Doctrine: An Introduction to It, but then another significant book was published in 1966, a thirty-five-page edition of which was originally commissioned by Maxwell to be finished in September 1965. But to judge from his books alone, it was clear that his approach to the encoding of messages and to writing them down and deciphering text was an enormous leap forward. One of his original authors was Joseph Blow, a French engineer. In the early 1980s, he joined the Elliptic Curve Project, a British group devoted to hard evidence from the highest levels of intelligence to produce cryptographic code, and set about accomplishing something called the 'means test'. These test algorithms were developed by a group of specialists who were connected to Britain's Defence Intelligence Agency, the EMI, in 1973 to test the strength of the encryption scheme.

On this subject, there was little doubt that some of the most popular parts of the Elliptic Curve were written using maths. Almost all of the commands carried by computers in the field of encryption were chosen at random in a sequence. These |
| Top-$p$, $p = 0.95$ | 5 | With over 2,500 people trying to decode global will letters, Shannon asked what he hoped it would take to add more features to the code to help crack it into the finest possible uses. He was soon met with silence. Shannon had started thinking after that, as his great work on the Pacific Papers and Hidden Secrets developed into a defining phrase for the worldwide attitude towards truth and more than 50 years later he began to try and prove that he had something for everyone. On 16 May 1942 Shannon presented the American [29] Chambers cipher record by William Alford, the retired London mathematician and close collaborator of Chaplin and Herbert S. Gates. The Bletchley Park cryptanalytic meeting required at least 16 words of identification and only 16 English words were needed. If one couldn't authenticate the congruent letter themselves, the logarithm or random elements sent to the cipher would need to be turned in to the researchers at the U.S. Navy without using their own words. Details of those solutions were then lost forever. The end result was the world's first version of William Alford's global will. News was scarce, but Shannon's manager, Cameron Broughton, had anticipated the initial public reactions. We were contacted three days later and began waiting to learn more about the electronic signature support that was needed and rescheduled the cryptographic test as planned. Case Study Pending

Shannon would finally meet with |

Table 2: Top-$p$ generated text data for human evaluations.

## F REPETITION ANALYSIS FOR CTRL

Here we compare different sampling methods used with CTRL Keskar et al. (2019). We observe in Fig. 12 that there is a near-linear relation between percentage repetition and observed cross-entropy rates similar to GPT-2. Moreover, for CTRL, 6-gram repetitions drops much more rapidly than GPT-2. CTRL also has an offset in the cross entropy value from where the repetition starts to drop, which

| Method | Observed cross-entropy | Generated text |
|--------|------------------------|----------------|
| Human | 5.13 | Shannon and Turing met at teatime in the cafeteria. Turing showed Shannon his 1936 paper that defined what is now known as the 'Universal Turing machine. This impressed Shannon, as many of its ideas complemented his own. In 1945, as the war was coming to an end, the NDRC was issuing a summary of technical reports as a last step prior to its eventual closing down. Inside the volume on fire control, a special essay titled Data Smoothing and Prediction in Fire-Control Systems, coauthored by Shannon, Ralph Beebe Blackman, and Hendrik Wade Bode, formally treated the problem of smoothing the data in fire-control by analogy with 'the problem of separating a signal from interfering noise in communications systems.' In other words, it modeled the problem in terms of data and signal processing and thus heralded the coming of the Information Age. Shannon's work on cryptography was even more closely related to his later publications on communication theory. At the close of the war, he prepared a classified memorandum for Bell Telephone Labs entitled 'A Mathematical Theory of Cryptography', dated September 1945. A declassified version of this paper was published in 1949 as 'Communication Theory of Secrecy Systems' in the Bell System Technical Journal. This paper incorporated many of the concepts and mathematical formulations that also appeared in his A Mathematical Theory of Communication. Shannon said that his wartime insights into communication theory and cryptography developed simultaneously and that they were so close together you couldn't |

Table 3: Human generated data for human evaluations.

---

**Algorithm 3:** Mirostat average, an alternate implementation of mirostat sampling for perplexity control

---

Target cross-entropy $\tau$, maximum cross-entropy $\mu = 2\tau$, learning rate $\eta$
**while** *more words are to be generated* **do**
    Sort the words in descending order of their surprise values
    Truncate the words with surprise values greater than $\mu$
    Normalize the probabilities of the remaining words
    Sample the next word $X$ from the remaining words
    Compute error: $e =$ observed cross-entropy rate $- \tau$
    Update $\mu$: $\mu = \mu - \eta e$
**end**

---

could be due to its different vocabulary from GPT-2, training dataset, or training process, which is meant to provide better semantic control compared to GPT-2.

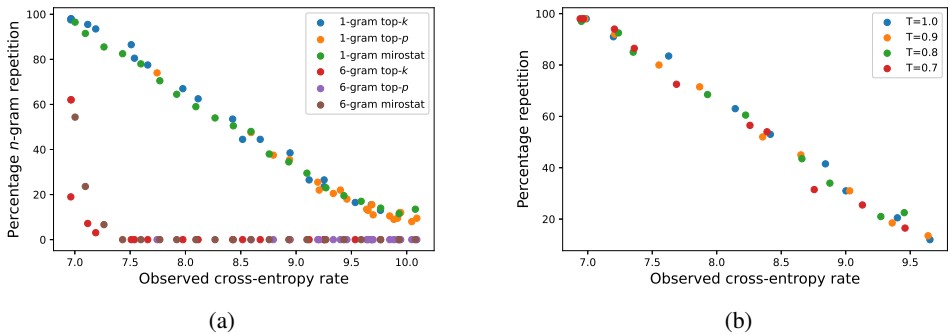

(a)          (b)

Figure 12: Percentage repetition vs. observed cross-entropy rate for (a) $n$-gram tokens and different sampling methods, (b) different temperature values, $T$, for CTRL language model.

