# OpenReview forum: "MIROSTAT: A NEURAL TEXT DECODING ALGORITHM THAT DIRECTLY CONTROLS PERPLEXITY"
_ICLR.cc/2021/Conference — ICLR 2021 Poster_

### Official Review · AnonReviewer1 · 2020-10-28
**Official Blind Review #1**

**Rating:** 7
**Confidence:** 4

**Review:**

Summary:

Neural text generation models typically rely on sampling schemes for autoregressive decoding. This may range from pure sampling, top-k, top-p to temperature modulated sampling. These methods are mostly heuristic schemes and lack theoretical analysis. This paper tries to fill that gap by analyzing these schemes theoretically under the Zipfian distribution assumption (an underlying distribution in natural language corpora and generally true for open-ended language generation models). While filling the theoretical gaps, this work proposes an adaptive top-k decoding mechanism - Mirostat. This is based on the understanding that cross-entropy is a useful measure of the quality of the generated text.

Strengths:

1. Theoretical analysis of the previously proposed sampling schemes in terms of surprise, cross-entropy, perplexity.
2. Adaptive top-k algorithm based on updating the exponents in Zipfian distribution, per time step.
3. Clarity of the paper.

Weaknesses:

1. It seems like a time complex scheme for decoding based on the formulation
2. Lacks generated samples in the main paper for qualitative assessment.
3. Human evaluation missing.

Questions:

1. Current neural text generation models are trained to optimize the cross-entropy loss. In an ideal scenario lower the cross-entropy (hence perplexity), the better should be the samples. Since this work (as well as its corresponding related works) show that humans don't form sentences that have the lowest perplexity (when measured using the trained model), does this work help in highlighting that cross-entropy loss might not be the correct objective for optimization?
2. In Algo 1: Do you first perform top-m for obtaining s and then perform top-k later based on the value obtained? If so, what is the time complexity of this process, as a function of m?
3. From Fig 4(c), it seems that humans prefer sentences having cross-entropy rates ~ 5.25 on avg. Why do you suggest 3 as the target average surprise?
4. The point about ad-hoc settings being applied for top-p and not for mirostat seems weird. This paper suggests 3.0 as the setting for target average surprise (based on empirical data) while top-p suggests ~0.9 as the setting for p (again based on empirical data - and shown in this paper that CE remains more or less constant wrt the number of generated tokens at p=0.9). In the end, a certain bit of ad-hoc tuning needs to be made for both cases. Why would this algorithm be better than nucleus sampling (Given it is more time consuming)?

Overall:

a) The theoretical analysis helps in understanding why top-k and top-p perform the way they do.

b) Based on the underlying assumptions:
1. Zipfian distribution
2. cross-entropy loss optimization
3. Lowest perplexity does not always mean good quality generation.

(I somehow feel a lot of disconnect between assumption 2 and 3), this work proposes an algorithm for adaptive top-k decoding to produce arguably "better" samples.

Please let me know if I am misunderstanding something. I am open to revising my assessment based on the answers to the questions raised.

-----
Update:

Updated score based  on the author response. That being said, it would still be good to see a plot of time complexity of the decoding scheme as a function of m.

---

> ### Author Response · Authors · 2020-11-18
> **Human experiments and detailed explanations**
>
> Updates and comments on weaknesses:
>
> 1. It is in fact a very efficient scheme with constant computational overhead on top-k, which is negligible in practice.
> 2. We have shifted these samples to the appendix for space constraint.
> 3. We have now added human evaluations in Sec.5.4.
>
> On Questions:
>
> 1. Cross-entropy is a function of two probability distributions. While training a language model, the goal is usually to minimize the cross-entropy between human probability distributions and language model distributions. This work is concerned with text decoding which is done post optimization. Here, the cross-entropy is computed between the trained language model and distorted language model.
> During the training process, the idea is to reduce the distance between human language distribution and language model distribution. Now, for computing the distance measure, cross-entropy is used. This work is not concerned with the distance measure itself; rather, this work is concerned with finding a good way to sample from a pretrained language model. Hence, this work does not directly highlight anything about the training process.
> The motivation for this work is similar to top-k or top-p decoding, where we want to truncate the unreliable tail of the language model. Here, the method of truncating actively observes the generated text and statistically guides it throughout the generation process in a way that maintains the observed cross-entropy rate of the text.
> We have now added human experiments in Sec. 5.4 that show that human evaluation of fluency, coherence, overall quality, and human-like properties of the generated text are maximized for a value of cross-entropy around 3. This, in addition to the fact that existing sampling methods have fluctuations in the observed cross-entropy of their generated texts for fixed value of parameters encourages us to choose mirostat over other decoding algorithms.
>
> 2. In our algorithm, $m$ is fixed as 100 to estimate the value of $s$, which is the parameter used in Zipf’s distribution. This algorithm only has a constant computational overhead on top-$k$ sampling and hence is computationally efficient.
> We have also implemented an alternate algorithm mirostat 2.0 in Appendix E that also controls perplexity without any estimation. We show that this algorithm performs near-identically as mirostat on control of perplexity and repetitions.
> Hence, for this work, we have chosen to estimate a parameter $s$ with a constant number of probabilities and directly perform top-$k$ sampling using the computation. Hence, the computational overhead for this algorithm compared to top-k sampling is linear in $m$, which when set to a small constant like 100 is negligible in practice.
>
> 3. Cross-entropy rates of 5.25 and 3 for humans and our suggested target respectively measure the distance of these distributions from the language model. Note that equal distance of a distribution from language model does not imply the same distribution. Logically, one would want to generate text from distribution as close to the language as possible since the language model has been trained to be as much close as possible to the human language distribution. However, as observed in our experiments in Fig. 3 we find that for low target average surprise, there is high repetitions in the text, and, we find that there is negligible sentence-level repetitions when the target value is set greater than 3.
> This point can also be verified from our newly added human experiments in Sec. 5.4, which shows that the overall quality of the text is maximum for observed cross-entropy rate around 3, which is closest to the original language model while having no sentence level repetitions.
>
> 4. Although we have a tuning parameter like top-k and top-p, the main difference is that we are actively controlling an important statistic of the text generated by controlling this parameter, which is unlike top-k or top-p since these sampling algorithms set the parameter at the very start of the generation process and do not monitor the generation process.
> Further, in Sec. 5.4, we experiment with both top-p and mirostat sampling for human evaluations and find that for both methods, the maximum human ratings are when the observed cross-entropy rate is close to 3. Moreover, from Fig. 2.b. note that top-p has significant fluctuations in the observed cross-entropy rate of the generated text even for fixed value of p, thus can have varying quality of texts for fixed input parameters.
> Hence, even though we have tuning of a parameter involved, it can be tuned in a more principled manner and not entirely ad-hoc, which provides a better guarantee of generating high-quality texts when target surprise value is set at an appropriate value, compared to other methods, where such guarantees might not be possible because of much fluctuations in the statistics of the output.
>
>
> The disconnect between 2 and 3 can be explained by our responses to questions 1 and 3.

---

> > ### Author Response · Authors · 2020-11-24
> > **Thank you for the detailed review**
> >
> > We thank the reviewer for their positive review and interesting questions.

---

### Official Review · AnonReviewer4 · 2020-10-28
**Official Blind Review #4**

**Rating:** 6
**Confidence:** 3

**Review:**

What is this paper about, what contributions does it make, what are the main strengths and weaknesses?

The paper proposes a feedback-based adaptive top-k decoding algorithm, named mirostat, to solve the repetition and incoherence problems in text generation. The mirostat algorithm adjusts the value of k run-time in the top-k algorithm during the testing stage to control the perplexity of the generated text near the preset value. Experimental results demonstrate the effectiveness of the proposed algorithm.

The main strengths are

1)	The repetition problem is common in the field of open-ended text generation. The proposed method aims to address this issue by controlling the perplexity of the generated text, which is novel and interesting. According to the examples shown in the experimental results, the algorithm can improve the quality of the generated text.

2)	In the related work section, the classification and explanation of relevant research are detailed. Related tasks and existing solutions are compared, and the rationality of motivation is proved by the research conclusion of existing work.

3)	The experiments are clearly organized. Their results show the control effect of mirostat over the cross-entropy rate and the relationship between the cross-entropy rate and the repetition degree of the generated text. It is good proof that mirostat can reduce the repetition degree by controlling PPL.

The main weaknesses are

1)	This paper proposes that a moderate perplexity (neither too large nor too small) is the best for the quality of the generated text, which has not been fully proven in experiments. In the experiment, only the relationship between PPL and the repetition degree of the generated text is proved. However, the relationship between PPL and other dimensions in the quality evaluation, such as coherence and fluency, is not studied.

2)	Some methods to solve the repetition issue are listed in the related work section, including the methods in the training stage and the testing stage. Among them, the methods worked in the testing stage, such as variants of beam search and variants of top-k sampling, is not selected as the comparison, but only two weak baselines are selected in the experiment.

3)	The proposed decoding algorithm is only experimentally verified on the GPT-2 model and has not verified its universality under other generative models or other generative tasks.

4)     In the related work section, some works such as Li et al. (2016), Vijayakumar et al. (2018), and Kulikov et al. (2019) may concern more about "quality-diversity tradeoff" instead of "repetitions".

---

> ### Author Response · Authors · 2020-11-18
> **Added human evaluations, experiments with repetition penalty, a different language model**
>
> Response to weaknesses:
>
> - "quality evaluation, such as coherence and fluency, is not studied.": We have now added human evaluations in Sec. 5.4, where we show the importance of controlling observed cross-entropy rate in generating high-fluency, high-coherence, high-quality, and human-like texts. We find that observed cross-entropy of 3.0 maximizes all these metrics for GPT-2 with 117M parameters.
>
> - "only two weak baselines are selected in the experiment": We have now added experimental results with repetition penalty previously used in [1] for reducing repetitions in Sec. 5.2. We find that using repetition penalty during the testing process indeed reduces the repetitions for fixed cross-entropy rates as shown in Fig. 4.a. However, in Fig. 4.b, we see that the performance of repetition penalty is not very consistent in reducing repetitions, hence using repetition penalty might not be a reliable method in reducing repetitions. Also, note from Fig. 4 that there is a new near-linear relation between cross-entropy rate and percent repetition when repetition penalty is used. Hence, there is a possibility that mirostat when used with repetition penalty can make the method of repetition penalty more reliable in reducing percent repetitions. We leave that for future investigation.
>
> - "The proposed decoding algorithm is only experimentally verified on the GPT-2 model and has not verified its universality under other generative models or other generative tasks.": We have added experiments from another language model, CTRL [1], in Appendix F and find that there is a similar near-linear relation between cross-entropy rates and percent repetitions, however, as added in the paper, we find there are some offset in this linear relation, which could have resulted from different vocabulary, training set, or training process, which is meant to provide better semantic control compared to GPT-2.
>
> - "In the related work section, some works such as Li et al. (2016), Vijayakumar et al. (2018), and Kulikov et al. (2019) may concern more about "quality-diversity tradeoff" instead of "repetitions".": Thanks for pointing this out. We have shifted the references to the correct places.
>
> [1] Nitish Shirish Keskar, Bryan McCann, Lav R. Varshney, Caiming Xiong, and Richard Socher. CTRL: A conditional transformer language model for controllable generation. arXiv:1909.05858v2 [cs.CL]., September 2019.

---

> > ### Author Response · Authors · 2020-11-24
> > **Thank you for the detailed review**
> >
> > We thank the reviewer for pointing out the points of improvement and help us improve the overall quality of the work.

---

### Official Review · AnonReviewer2 · 2020-10-28
**Review: MIROSTAT: A NEURAL TEXT DECODING ALGORITHM THAT DIRECTLY CONTROLS PERPLEXITY**

**Rating:** 6
**Confidence:** 5

**Review:**

In the context of neural text generation, the authors study how perplexity varies with top-$k$ and top-$p$ sampling and propose a sampling algorithm that uses Zipf's law to dynamically adjust $k$ in order to control per-sequence perplexity.

Overall, the theoretical analysis and relationship between log probability and repetition was interesting, but there are several concerns with the method and experimental evaluation, detailed below. The idea is interesting and I hope the authors continue down this line, but in its current form I would not recommend acceptance. (edit: see discussion below, I have adjusted the score to above the acceptance threshold)


#### Pros
- The theoretical analysis of cross-entropy growth with top-$k$ versus top-$p$ was interesting (e.g. summarized in Figure 1).
- Nice empirical demonstration of repetition correlating with log probability.

#### Clarity
- The presentation in Section 2 could be simplified or made more concrete - overall it seems like this section is building up to a standard definition of perplexity in a complicated way.
    - are these generic definitions of cross-entropy rate and perplexity (defined using the Shannon-McMillan-Breiman theorem) needed? I don't see them used in the main text, so it would be helpful to shorten this section, or concretely say how each step corresponds to a language model.
    - Equation (3) assumes that $P_N$ is a stationary ergodic source. Why can a neural language model be considered a stationary ergodic source?
    - Why *surprise* instead of *information content*? *Surprise rate* is not used again in the text.
    - In the abstract you say "target value of perplexity" but then $\tau$ is called the "target surprise value", and in Appendix A it reports "target *average* surprise value". In this review I'll use 'perplexity', but it would be helpful to check whether there are inconsistencies in the paper.

#### Method

- **Dependence on hyperparameter.** The authors mention that for low values of $k$ and $p$, perplexity drops, leading to repetition, while for high values of $k$ and $p$, perplexity increases, leading to incoherence. The authors claim that *Mirostat avoids both traps*. However it requires setting a target value ($\tau$). What is the difference between having to choose $k$ or $p$ versus choosing $\tau$? Wouldn't Mirostat fall into the traps with low $\tau$ or high $\tau$ (e.g. Figure 4.d)? Since you showed that perplexity grows linearly with $p$ (Fig 1), why is Mirostat needed versus using top-$p$?

- **Fixed perplexity per continuation.** Mirostat enforces the average token log-probability of each *individual continuation* to be near a hyperparameter $\tau$. However, won't the "ideal" perplexity vary based on the prefix? I.e. for some prefixes there may be low conditional entropy in the true distribution, meaning a small number of high-probability continuations are much more reasonable than others. In this case, generating a sequence with perplexity based on $\tau$ would filter out these high-probability continuations. Could the authors comment on this issue? The underlying assumption of the method is that it is a good idea to have a fixed perplexity for all continuations.

- **Zipf's motivation.** While I understand that the Zipf's law assumption was needed to derive the theoretical results, it's unclear why Zipf's law is used to motivate the practical method (Algorithm 1). Why would we want to estimate the zipf's exponent on the top-100 words at each timestep, and choose k using (7)? This motivation, and a comment on the guarantees it gives us on full sequences, should be more clearly stated.

#### Experiments

- **Simple 'perplexity target baselines'.** Related to the "Zipf's motivation" comment above, what's missing is evaluating different ways of controlling the perplexity, in order to evaluate that the proposed method based on Zipf's law is the best (for some definition of best). If the goal is to control the perplexity of each sequence, why not sample several sequences with top-$k$ and choose one that has perplexity close to a target $\tau$? What is the performance of adjust $k$ or $p$ based on a different heuristic, e.g. absolute difference between the perplexity of the sequence-so-far and $\tau$?

- **Human evaluation.** Human evaluation is required to get a full measure of the generated text's quality; currently the paper just argues for quality/coherence by showing a few examples. It's possible that this form of dynamic $k$ adjustment introduces some artifacts. For instance, the behavior of the "surprise" in Figure 5.d under Mirostat doesn't resemble that of humans, so it's possible that some odd behavior is introduced.

- **Misc.** In figure 5, why is mirostat preferable to top-k or top-p? Why are $k,p$ of 0.4 and 1.0 and $\tau=1.8$ selected here?

---

> ### Author Response · Authors · 2020-11-18
> **Added human evaluations and alternate implementations**
>
> Clarity:
> A. We have rewritten this part in a concise paragraph form.
> For theoretical results, we work with cross-entropy for ease of analysis and for practical results we work with surprise rate for ease of computation. The two quantities are different unless connected using the stationary ergodic property of language [1]. We have added this reference in the paper, which was missing previously.
>
> B. Stationary ergodic assumption: We have added a standard textbook reference justifying the stationary ergodic model [1].
>
> C. Surprise vs information content: Information content, self-information, or surprisal all refer to the same quantity (–log{p}). We modified it to the term "surprise" to emphasize the conditional probability present in this term, which is not the case for surprisal.
>
> D. On inconsistencies: We have changed the term “target average surprise” to “target surprise” to make things consistent. We have retained “target value of perplexity” in the abstract since surprise is defined only later in the paper, but “perplexity” is a commonly-known term in NLP.
>
> Method:
>
> A. Dependence on hyperparameters: In mirostat, the parameter is chosen in a more principled way: it fixes a statistical property of the generated text by actively monitoring the generation process unlike previous methods that do not monitor the process actively. Based on the results in Sec. 5.2 and the newly added human evaluation in Sec. 5.4, one can choose $\tau$ in a more principled way producing high-quality texts.
> Fig. 4.d (now Fig. 5.d) shows that the observed cross-entropy rate converges to its target rate.
> Although, cross-entropy grows linearly with $p$, in practice, in Fig.2, there is much fluctuation in observed cross-entropy rate. From human evaluations in Sec. 5.4 we see that there is a quality difference between texts with different observed cross-entropy rates. Hence, we need better control of the statistics of the output text, which is why we need mirostat.
>
> B. Fixed perplexity per continuation: We agree that there might be better target surprise values (or varying target surprise values as a function of the length of text generated). Having feedback to control the average conditional probability to bring it to a fixed value is only a starting point.
> Finding better target values needs further experimentation and hence we leave it for future work. Notwithstanding, we are confident that for any such algorithm, controlling the generation process like in our work would be very important and that our work will form a good basis for any such work.
>
> C. Zipf's motivation: The estimation step only averages 100 values and then the value of k obtained can be used directly for top-k sampling. Hence, this algorithm has very little computational overhead as compared to top-k. Since we are just working with languages (like English) in our generation process that follow Zipf’s distribution, we chose this algorithm.
> We have added implementation of an algorithm, mirostat 2, that does not assume Zipf’s law, in Appendix E. In Fig. 10, we show that its performance is nearly identical to mirostat in terms of controlling perplexity and repetitions. But, note it has a varying additional time complexity depending on vocabulary size and target surprise value on top of top-k, unlike mirostat.
>
> Experiments:
>
> A. Simple 'perplexity target baselines': We have added new algorithms to control perplexity in Appendix E: mirostat 2 and mirostat-average. Mirostat 2 is designed in the same way as mirostat except that it no longer uses Zipf’s law. Mirostat-average has a heuristic of difference between $\tau$ and the observed cross-entropy rate of the sequence so far.
> We find mirostat 2 to work nearly identically to mirostat in terms of control and repetition, whereas mirostat-average performs poorly.
> Sampling several top-p samples and choosing the one with desired cross-entropy is difficult in practice due to the fluctuations in observed cross-entropy rate in top-p.
>
> B. Human evaluation: We have added human evaluations in Sec. 5.4 evaluating the fluency, coherence, quality, and human-likeness of the generated text for mirostat and top-p. For both mirostat and top-p, all the metrics are maximized for observed cross-entropy rate = 3 from the set {2.5,3,4,5}.
> Fig. 5.d (now Fig. 7.d) shows how mirostat avoids the boredom trap. Both Fig. 5.a (now Fig.7.a) that uses top-p with $p=0.4$ and Fig. 5.d (now Fig. 7.d) that uses mirostat with $\tau=1.8$ starts with a similar observed cross-entropy rate. Both fall into boredom traps after generating roughly 100 words, but mirostat comes out of it, whereas top-p sampling does not. Hence, indicating that mirostat avoids boredom trap.
>
> C. Misc: Same as the response to the previous question on the choice of the parameters.
>
> [1] Christopher Manning and Hinrich Schutze. Foundations of Statistical Natural Language Processing. MIT Press, 1999.

---

> > ### Comment · AnonReviewer2 · 2020-11-23
> > **Thanks for the updates and responses**
> >
> > Thanks for the detailed response and the updates. With the clarified terms, the additional baselines, and the human evaluation, you have addressed the major points in the review. The paper makes interesting contributions and can serve as a base for further work, so I'm updating my score to above the acceptance threshold.

---

> > > ### Author Response · Authors · 2020-11-24
> > > **Thanks for the detailed review**
> > >
> > > We thank the reviewer for their detailed review and help improve the contribution of the paper.

---

### Decision · Program_Chairs · 2021-01-07
**Final Decision**

**Decision:**

Accept (Poster)

**Comment:**

This work presents a novel approach to improving text decoding. This is backed up by a solid analysis of cross-entropy growth with top-k vs top-p and an interesting demonstration of repetition correlating with probability. The paper is well written and well organized. The authors' rebuttal was effective in convincing the reviewers. The human evaluation (added during the rebuttal phase) is a good demonstration of the effectiveness of the approach and so this paper's proposed decoding algorithm is likely to be impactful.

Pros:
- Well written.
- Solid theoretical analysis of cross-entropy and its relation to top-p and top-k decoding. Good demonstration of how repetition is related to probability.
- Interesting, novel and effective decoding algorithm.
- Human evaluation of the algorithm's output.

Cons:
- The approach has not been tested with a variety of language models.
- Decoding quality still depends on a target perplexity which may need to be tuned.
- Unnecessary dependence on Zipf's law in the basic decoding algorithm.